# A Consciousness-Inspired Planning Agent for Model-Based Reinforcement Learning

**Mingde Zhao**[1,4,*], **Zhen Liu**[2,4,*], **Sitao Luan**[1,4,*], **Shuyuan Zhang**[1,4,*]
**Doina Precup**[1,3,4,5†], **Yoshua Bengio**[2,4,5†]
[1]McGill University; [2]Université de Montréal; [3]DeepMind; [4]Mila; [5] CIFAR AI Chair
[*]: Equal Contribution, [†]: Equal Supervision

## Abstract

We present an end-to-end, model-based deep reinforcement learning agent which dynamically attends to relevant parts of its state during planning. The agent uses a bottleneck mechanism over a set-based representation to force the number of entities to which the agent attends during planning to be small. In experiments, we investigate the bottleneck mechanism with several sets of customized environments featuring different challenges. We consistently observe that the design allows the planning agents to generalize their learned task-solving abilities in compatible unseen environments by attending to the relevant objects, leading to better out-of-distribution generalization performance. Check project page `https://github.com/PwnerHarry/CP`.

## 1 Introduction

Whether when planning our paths home from the office or from a hotel to an airport in an unfamiliar city, we typically focus on a small subset of relevant variables, *e.g.* the change in position or the presence of traffic. An interesting hypothesis of how this path planning skill generalizes across scenarios is that it is due to computation associated with the conscious processing of information [2, 3, 14]. Conscious attention focuses on a few necessary environment elements, with the help of an internal abstract representation of the world [43, 14]. This pattern, also known as consciousness in the first sense (C1) [14], has been theorized to enable humans' exceptional adaptability and learning efficiency [2, 3, 14, 43, 7, 15]. A central characterization of conscious processing is that it involves a *bottleneck*, which forces one to handle dependencies between very few environmental characteristics at a time [14, 7, 15]. Though focusing on a subset of the available information may seem limiting, it facilitates Out-Of-Distribution (OOD) and systematic generalization to other situations where the ignored variables are different and yet still irrelevant [7, 15].

In this paper, we encode some of these ideas into reinforcement learning agents. Reinforcement learning (RL) is an approach for learning behaviors from agent-environment interactions [41]. However, most of the big successes of RL have been obtained by deep, model-free agents [30, 37, 38]. While Model-Based RL (MBRL) has generated significant research due to the potentials of using an extra model [31], its empirical performance has typically lagged behind, with some recent notable exceptions [36, 24, 17].

Our proposal is to take inspiration from human consciousness to build an architecture which learns a useful state space and in which attention can be focused on a small set of variables at any time, where the aspect of "partial planning"[1] is enabled by modern deep

---

[1]Partial planning is interpreted in different ways. For example, concurrent work [26] focuses on modelling "affordable" temporally extended actions, *s.t.* an "intent" could be achieved more efficiently.

35th Conference on Neural Information Processing Systems (NeurIPS 2021).

RL techniques [42, 26]. Specifically, we propose an end-to-end latent-space MBRL agent which does not require reconstructing the observations, as in most existing works, and uses Model Predictive Control (MPC) framework for decision-time planning [34, 35]. From an observation, the agent encodes a set of objects as a state, with a selective attention bottleneck mechanism to plan over selected subsets of the state (Sec. 4). Our experiments show that the inductive biases improve a specific form of OOD generalization, where consistent dynamics are preserved across seemingly different environment settings (Sec. 5).

## 2 Background & Context

We consider an agent interacting with its environment at discrete timesteps. At time $t$, the agent receives observation $o_t$ and takes action $a_t$, receiving a reward $r_{t+1}$ and new observation $o_{t+1}$. The interaction is episodic. The agent is also building a latent-space transition model, $\mathcal{M}$, which can be used to sample a next state, $\hat{s}_{t+1}$, a reward $\hat{r}_{t+1}$ and a binary signal $\hat{\omega}_{t+1}$ which indicates if the model predicts termination after the transition. We will now compare and contrast our approach with some existing methods from the MBRL literature, explaining the rationale for our design choices.

**Observation Level Planning and Reconstruction *vs* Latent Space Planning**
Many MBRL methods plan in the observation space or rely on reconstruction-based losses to obtain state representations [24, 36, 17, 48]. Appropriate as these methods may be for some robotic tasks with few sensory inputs, *e.g.* continuous control with joint states, they are arguably difficult with high-dimensional inputs like images, since they may focus on predictable yet useless aspects of the raw observations [31]. Besides suffering from the need to reconstruct noise or irrelevant parts of the signal, it is not clear if representations built by a reconstruction loss (*e.g.* $L_2$ in the observation space) are effective for an MBRL agent to plan or predict the desired signals [39, 17, 18], *e.g.* values (in the RL sense), rewards, *etc.*. In this work, we use an approach similar to those in [39, 36, 17], building a latent space representation that is jointly shaped by all the relevant RL signals (to serve value estimation and planning) without using reconstruction.

**Staged Training *vs* End-to-End Training**
Some MBRL agents based on a world model [16, 24, 31] use two explicit stages of training: (1) an inner representation of the world is trained using exploration (usually with random trajectories); (2) the representation is fixed and used for planning and MBRL. Despite the advantages of being more stable and easier to train, this procedure relies on having an environment where the initial exploration provides transitions that are sufficiently similar to those observed under improved policies, which is not the case in many environments. Furthermore, the learned representation may not be effective for value estimation, if these transitions do not contain reward information that can be used to update the input-to-representation encoder. End-to-end MBRL agents, *e.g.* [39, 36], are able to learn the representation online, simultaneously with the value function, hence adapting better to non-stationarity in the transition distribution and rewards.

**Type of planning**
MBRL agents can use the model in different ways. Dyna [40] learns a model to generate "imaginary" transitions, which contribute to the training of the value estimator [40], in addition to the real observations, thus boosting sample efficiency. However, if the model is inaccurate, the transitions it generates may be "delusional", which may alter the value estimator and negatively impact performance. Moreover, Dyna is typically used to generate extra transitions from the states visited in a trajectory, and updates the model based on the observed transitions as well. This means Dyna is focused on the data distribution encountered by the agent and may have trouble generalizing OOD. In contrast, simulation-based model-predictive control (MPC) and its variants [34, 35, 18] only update the value estimator based on real data, using the model simply to perform lookahead at decision-time. Hence, model inaccuracies impact less, with more favorable OOD generalization capabilities. Hence, MPC is adopted in our approach.

**Vectorized *vs* Set Representations for RL**
Most Deep Reinforcement Learning (DRL) work focus on learning vectorized state rep-

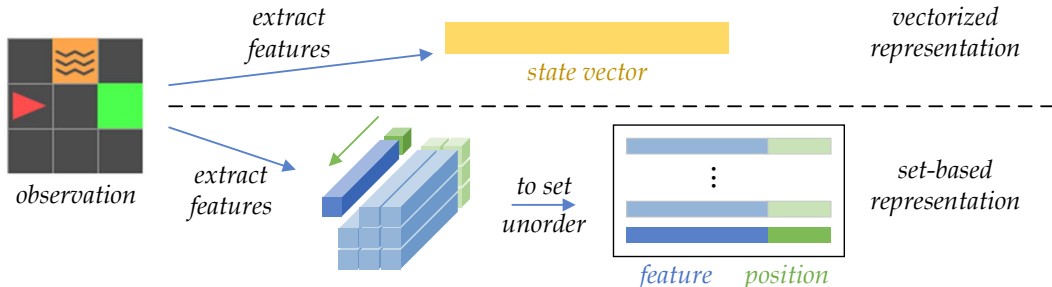

Figure 1: **Set-based state encoder** compared to classical vectorized state encoders: the feature map extracted by some feature extractor, *e.g.* a CNN, is "chopped" into feature vectors and concatenated with positional information. All of the resulting concatenations are treated as *objects* in a set, capturing the features of observed entities. The permutation-invariance of set computations forces the learner to be robust to small changes in the set (*e.g.* one of the elements being different or missing).

resentations, where the agents' observation is transformed into a feature vector of fixed dimensionality [30, 19]. Instead, set-based encoders, *a.k.a.* object-oriented architectures, are designed to extract a set of unordered vectors from which to predict the desired signals via permutation-invariant computations [50], as illustrated in Fig. 1. Recent works in RL have shown the promise of set-based representations in capturing environmental states, in terms of generalization, as well as their similarities to human perception [13, 47, 32, 46, 29]. Additionally in this work, we utilize the compositionality of set representations to enable the discovery of sparse interactions among objects, *i.e.* underlying dynamics, as well as to facilitate the bottleneck mechanism, analogous to C1 selection. The set-based representation coupled with the bottleneck provides an inductive bias consistent with selecting only the relevant aspects of a situation on-the-fly through an attention mechanism. The small size of the working memory bottleneck also enforces sparsity of the dependencies [7, 15] captured by the learned dynamics model: each transition can only relate a few objects together, no more than the size of the bottleneck.

## 3 MBRL with Set Representations

We present an end-to-end baseline MBRL agent that uses a set-based representation and carries out latent space planning, but **without** a consciousness-inspired small bottleneck. This agent serves as a baseline to investigate the OOD generalization capabilities brought by the bottleneck, which is to be introduced later in Sec. 4.

The mapping from observations to values is a combination of an *encoder* and a *value estimator*. The encoder maps an observation vector to a set of objects, which constitutes the latent state. The value estimator is a permutation-invariant set-to-vector architecture that maps the latent state to a value estimate. Note that the same state set is used for all the agents' predictions, including future states, rewards *etc.*, as we will discuss later.

**Encoder.** For image-based observations, we use the features at each position of the CNN output feature map to characterize the feature of an object, similar to [9], as shown in Figure 1. To recover positional information lost during the process, we concatenate each object feature vector with a positional embedding to form a complete object embedding. Such approach is different from the common practice of mixing positional information by addition [45]. This is for the compatibility with our dynamics model training procedure, discussed below.

**(State-Action) Value Estimator** takes the form $Q : \mathcal{S} \to \mathbb{R}^{|\mathcal{A}|}$, where $\mathcal{S}$ is the learned state space by the set-based encoder (hoping to capture the real underlying state space of the MDP) and $\mathcal{A}$ is a discrete action set. We use an improved architecture upon DeepSets [50], depicted in Figure 2. The architecture performs reasoning on a set of encoded objects, resembling pervasive usage in natural language processing, where the objects are typically word tokens [33].

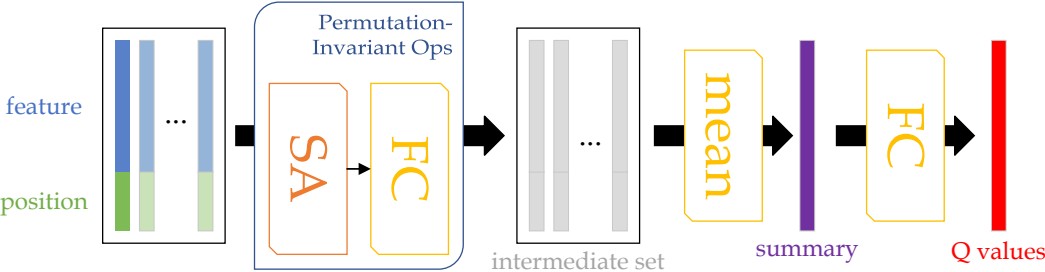

Figure 2: **Value estimator** $Q$ and **a generic set-to-vector architecture**: we modify the design of DeepSets [50] by replacing the MLP before pooling with transformer layers (multi-head Self-Attention (SA) + object-wise Fully Connected (FC)) [45]. We found this change to be helpful for performance. After applying the transformer layers, the intermediate set (colored gray) entangles features and positions. Please check the Appendix for more details on the self-attention operations involved.

**Transition Model.** The transition model maps from $s_t, a_t$ to $\hat{s}_{t+1}, \hat{r}_t$ and $\hat{\omega}_{t+1}$. We separate this into: 1) the **dynamics model**, in charge of simulating how the state would change with the input of $a_t$ and 2) the **reward-termination estimator** which maps $s_t, a_t$ to $\hat{r}_t$ and $\hat{\omega}_{t+1}$.

While designing reward-termination estimator is straightforward (a two-headed augmented architecture similar to the value estimator), the dynamics model requires regression on *unordered* sets of objects (set-to-set). A common approach is to use matching methods, *e.g.* Chamfer matching or Hausdorff distance, However, they are computationally demanding and subject to local optima [5, 8, 28]. Targeting this, our feature-position separated set encoding not only makes the permutation-invariant computations position-aware, but also allows simple end-to-end training over the dynamics. By forcing the positional tails to be *immutable* during the computational pass, we can use them to solve the matching trivially: objects "labeled" with the same positional tail in the prediction $\hat{s}_{t+1}$ (output of the dynamics model) and the training sample $s_{t+1}$ (state obtained from the next observation) are aligned, forming pairs of objects with changes *only* in the feature, as shown in Figure 3.

**Tree Search MPC.** The agent employs a tree-search based behavior policy (with $\epsilon$-greedy exploration). During planning, each tree search call maintains a priority queue of branches to simulate with the model. When a designated budget (*e.g.* number of steps of simulation) is spent, the agent greedily picks the immediate action that leads to the most promising path. We present the pseudocode of the Q-value based prioritized tree-search MPC in Appendix.

Equivalence could be drawn from this planning approach to Monte-Carlo Tree Search (MCTS) [37, 38]. While this method is far more simplistic and require fewer simulations for each planning call (see example in Appendix).

**Training.** The proposed agent is trained from sampled transitions with the following losses:

- Temporal Difference (TD) $\mathcal{L}_{\text{TD}}$: regresses the current value estimate to the update target, *e.g.* calculated according to DQN or Double DQN (DDQN) [30, 44]. In experiments, a distributional output is used for both value and reward estimation, making this loss a KL-divergence [6].

- Dynamics Consistency $\mathcal{L}_{\text{dyn}}$: A $L_2$ penalty established between the aligned $\hat{s}_{t+1}$ and $s_{t+1}$, where $\hat{s}_{t+1}$ is the imagined next (latent) state given $o_t, a_t$ and $s_{t+1}$ is the true next (latent) state encoded from $o_{t+1}$.

- Reward Estimation $\mathcal{L}_r$: the KL-divergence between the imagined reward $\hat{r}_{t+1}$ predicted by the model and the true reward $r_{t+1}$ of the observed transition.

- Termination Estimation $\mathcal{L}_\omega$: the binary cross-entropy loss from the imagined termination $\hat{\omega}_{t+1}$ to the ground truth $\omega_{t+1}$, obtained from environment feedback.

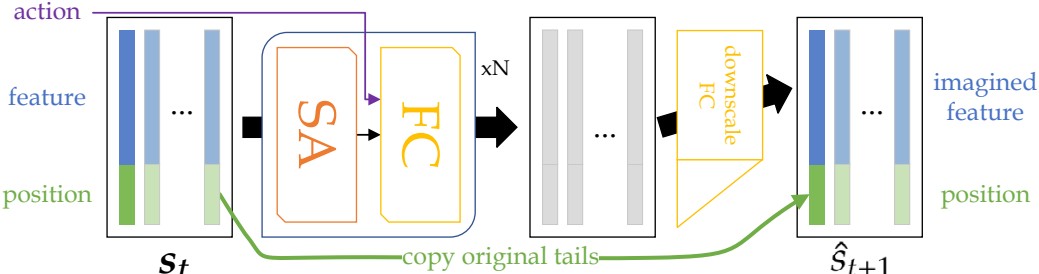

Figure 3: **Dynamics model**: for FC sub-layers of the transformer layers, we inject an action embedding *s.t.* the transformer computations are now action conditioned. After getting the intermediate set, we downscale each of the objects, leaving the positions untouched and directly copied from the input $s_t$. FC downscale is a linear transformation which downscales the dimensionality of the intermediate objects to that of the features part of objects (before the layernorm). In this way, after concatenating the positional tails the objects have consistent dimensionality. Intuitively, each object slot recovers its positional tail at the output. Though the objects in the sets (input-intermediate-output) are aligned, within each set they are still unordered, *i.e.* permutation-invariant.

The resulting total loss for end-to-end training of this set-based MBRL agent is thus[2]:

$$\mathcal{L} = \mathcal{L}_{\text{TD}} + \mathcal{L}_{\text{dyn}} + \mathcal{L}_r + \mathcal{L}_\omega$$

Jointly shaping the states avoids the representation collapsing to trivial solutions and makes the representation useful for all signal predictions of interest.

## 4   Consciousness-Inspired Bottleneck

In this section, we introduce an inductive bias which facilitates C1-capable planning. In a nutshell, the planning is expected to focus on the parts of the world that matter for the plan. Simulations and predictions are all expected to be performed on a (small) bottleneck set, which contains all the important transition-related information. As illustrated in Figure 4, the model performs 1) selection of the bottleneck set from the full state-set, 2) dynamics simulation on the bottleneck set and 3) integration of predicted bottleneck set to form the predicted next state.

**Conditional State Selection** We select a bottleneck set $c_t$ of $n$ objects from the potentially large state set $s_t$ of $m \gg n$ objects. Then we only model the transition for the selected objects in $c_t$. To make this selection, we use a key-query-value attention mechanism, where the key and the value for each object in $s_t$ are obtained from that object, and the query is a function of some learned dedicated set of vectors and of the action considered (see Appendix for details). Inspired by the work on self-attention for memory access [25], we use a semi-hard top-$k$ attention mechanism to facilitate the selection of the bottleneck set. That is, after the query, the top-$k$ attention weights are kept, all others are set to 0, and then the attention weights are renormalized. This semi-hard attention technique limits the influence of the ill-matched objects on the bottleneck set $c_t$ while allowing for a gradient to propagate on the assignment of relative weight to different objects. With purely soft attention, weights for irrelevant objects are never 0 and learning to disentangle objects may be more difficult.

**Dynamics / Reward-Termination Prediction on Bottleneck Sets.** We use the same architecture as described in Sec. 3, but taking the bottleneck objects as input rather than the full state set. Details of the architecture are in the Appendix.

**Change Integration.** An integration operation, intuitively the inverse operation of selection, is implemented to 'soft paste-back' the changes of the bottleneck state onto the state set $s_t$,

---

[2]In our experiments, no re-weighting is used for each term of the total loss. This is possible for the fact that they are in similar magnitudes. In our experimental implementation, no recurrent mechanism is used however the same training procedure is naturally extendable.

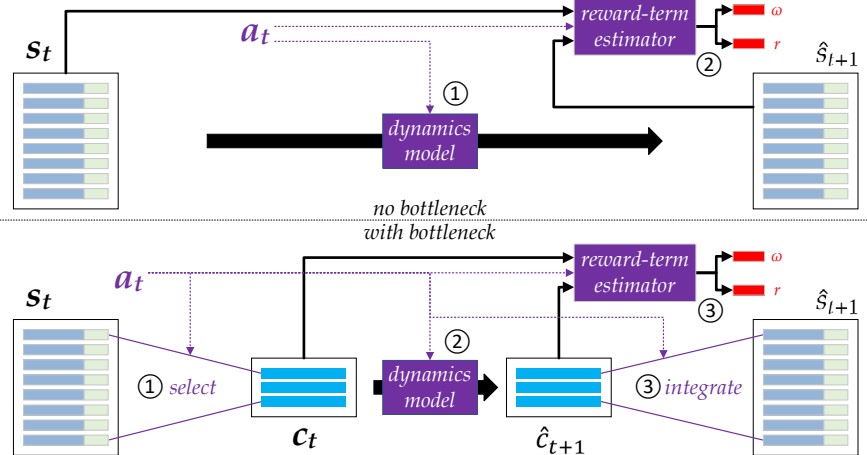

Figure 4: **Bottleneck stages** (operations colored in purple are conditioned on a chosen action): 1) a bottleneck set $c_t$ is soft-selected from the whole state (object set) $s_t$ through semi-hard multi-head attention; 2) dynamics are applied to the bottleneck set $c_t$ to form $\hat{c}_{t+1}$; 3) the reward and termination signals are predicted from $c_t$, $\hat{c}_{t+1}$ and $a_t$. Then, the changes introduced in $\hat{c}_{t+1}$ are integrated with $s_t$ to obtain $\hat{s}_{t+1}$, the imagined next state, with the help of attention. Note that the two computational flows in stage 3 are naturally parallelizable.

yielding the imagined next state set $\hat{s}_{t+1}$. This is also achieved by attention operations, more specifically querying $\hat{c}_{t+1}$ with $s_t$, conditioned on the action $a_t$. Please check the Appendix for more details.

**Discussion.** The bottleneck described in this section is a natural complement to the MBRL model with set representations discussed previously. In particular, planning and training are carried out the same way as discussed in Sec. 3.

We expect the Conscious Planning (CP) agent to demonstrate the following advantages:

- Higher Quality Representation: the interplay between the set representation and the selection / integration forces the representation to be more disentangled and more capable of capturing the locally sparse dynamics.

- More Effective Generalization: only essential objects for the purpose of planning participate in the transition, thus generalization should be improved both in-distribution and OOD, because the transition does not depend on the parts of the state ignored by the bottleneck.

- Lower Computational Complexity: directly employing transformers to simulate the full state dynamics results in a complexity of $O(|s_t|^2 d)$, where $d$ is the length of the objects, due to the use of Self-Attention (SA), while the bottleneck lowers it to $O(|s_t||c_t|d)$.

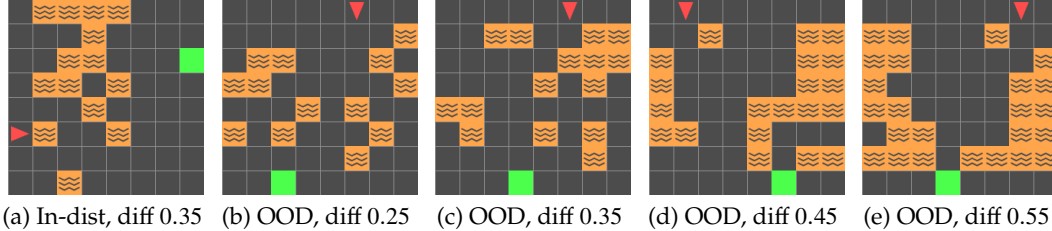

(a) In-dist, diff 0.35 (b) OOD, diff 0.25 (c) OOD, diff 0.35 (d) OOD, diff 0.45 (e) OOD, diff 0.55

Figure 5: **Non-Static RL Setting, with in-distribution and OOD tasks**: (a) example of training environments (b - e) examples of OOD environments (rotated 90 degrees, changing the distribution of grid elements). For OOD testing, we evaluate different levels of difficulty (b - e). The agent (red triangle) points in the forward movement direction. The goal is marked in green. For each episode (training or OOD), we randomly generate a new world from a sampling distribution. Note that the training environments and the OOD testing environments have no intersecting observations.

# 5 Experiments

We present our experimental settings and ablation studies of our CP agent against baselines to investigate the OOD generalization capabilities enabled by the C1-inspired bottleneck mechanism. To clarify, the OOD generalization we refer to specifically is *the agents' ability to generalize its learned task skills across seemingly different tasks with common underlying dynamics*. Take the set of experiments in this section for example, we want the agent to be able to generalize its navigation skills in unseen environments.

## 5.1 Environment / Task Description

We use environments based on the MiniGrid-BabyAI framework [11, 10, 21], which can be customized for generating OOD generalization tests with varying difficulties. To make sure we assess the agents as clearly as possible, the customized environments feature clear object definitions, with well-understood underlying dynamics based on object interactions. Furthermore, the environments are solvable by Dynamic Programming (DP) and can be easily tuned to generate OOD evaluation tasks. These characteristics are **crucial** for the experimental insights we are seeking.

In this section, the experiments are carried out on $8 \times 8$ gridworlds[3], as shown in Figure 5. The agent (red triangle) needs to navigate (by turning left, right or stepping forward) to the goal while dodging the lava cells along the way[4]. If the agent steps into lava (orange square), the episode terminates immediately with no reward. If the agent successfully reaches the goal (green square), it receives a reward of +1 and the episode terminates. For better generalization, the agent needs to understand how to avoid lava in general (and not at specific locations, since their placement changes) and to reach the goal as quickly as possible[5]. The environments provide grid-based observations that are ready to be interpreted as set representations: each cell of the observation array is an object, thus resulting in a set of 64 objects in $s_t$ for each observation.

For the agent to be able to *understand* the environment dynamics instead of *memorizing* specific task layouts, we generate a new environment for each training or evaluation episode. In each training episode, the agent starts at a random position on the leftmost or rightmost edge and the goal is placed randomly somewhere along the opposite edge. In between the two edges, the lava cells are randomly generated according to a *difficulty* parameter which controls the probability of placing a lava cell at each valid position. The difficulty parameter controls partially how seemingly different the OOD evaluation tasks are to the in-distribution training tasks, though we know the underlying dynamics of all these tasks are the same. For training episodes, the difficulty is fixed to 0.35. We note that most usual RL benchmarks contain fixed environments, where the agent is expected to acquire a specific optimal policy. These environments are ill-suited for our purpose.

For OOD evaluation, the agent is expected to adapt in new tasks with the **same** underlying dynamics in a 0-shot fashion, *i.e.* with the agent's parameters fixed. The OOD tasks are crafted to include changes both in the support (orientation) and in the distribution (difficulty): the agent is deployed in *transposed* layouts[6] with varying levels of difficulty ($\{0.25, \mathbf{0.35}, 0.45, 0.55\}$). The differences of in-distribution (training) and OOD (evaluation) environments are illustrated in Figure 5.

## 5.2 Agent Setting

We build all the set-based MBRL agents included in the evaluation on a common model-free baseline: a set-based variant of Double-DQN (DDQN) [44] with prioritized replay and distributional outputs. For more details, please check the Appendix.

---

[3]We provide additional results for world sizes ranging from $6 \times 6$ to $10 \times 10$ in the Appendix. $8 \times 8$ is chosen as the demonstrative case.

[4]In the Appendix, we provide additional test settings with different dynamics, which also demonstrates the agents' ability to work well despite cluttering distractions.

[5]Please check the Appendix for extra sets of tasks with different agent actions and task objectives.

[6]The agent starts at the top or bottom edge and the goal is respectively on the bottom or top edge, whereas a training environment has the agent and goal on the left or right edges

We compare the proposed approach, labelled CP in the figures (for Conscious Planning) against the following methods:

- *UP* (for Unconscious Planning): the agent proposed in Section 3, lacking the bottleneck.
- *model-free*: the model-free set-based agent is the basis for the set-based model-based agents. It consists of only the encoder and the value estimator, sharing their architectures with CP and UP.
- *Dyna*: the set-based MBRL agent which includes a model-free agent and an observation-level transition model, *i.e.* a transition generator. For the model, we use the CP transition model (with the same hyperparameters as the best performing CP agent) on the original environment features without an encoder. We also use the same hyperparameters as in the CP model training. The agent essentially doubles the batch size of the model-free baseline by augmenting training batches with an equal number of generated transitions.
- *Dyna\**: A Dyna baseline that uses the true environment model for transition generation. This is expected to demonstrate Dyna's performance limit.
- *WM-CP*: A world model CP variant that differs by following a 2-stage training procedure [16]. First, the model (together with the encoder) is trained with $10^6$ random transitions. After this, the encoder and the model are fixed and RL begins.
- *NOSET*: A UP-counterpart with vectorized representations and no bottleneck mechanism.

Particularly, for CP and UP agents, we also test the following variants:

- *CP-noplan*: A CP agent that trains normally but does not plan in OOD evaluations, *i.e.* carrying out model-free behavior. This baseline aims to demonstrate the impact of planning in the training process on the OOD capability of the value estimator.
- *UP-noplan*: UP counterpart of CP-noplan.

Note that the compared methods share architectures as much as possible to ensure fair comparisons. Details of the compared methods, their design and hyperparameters are provided in the Appendix.

## 5.3 Performance Evaluation

### 5.3.1 In-Distribution

In Figure 6, we present the in-distribution evaluation curves for the different agents. For UP, CP and the corresponding model-free baselines, the performance curves show no significant difference, which demonstrates that these agents are effective in learning to solve the in-distribution tasks. During the "warm-up" period of the WM baseline, the model learns a representation that captures the underlying dynamics. After the warm-up, the encoder and the model parameters are fixed and only the value estimator learns to predict the state-action values based on the given representation. The increase in performance is not only delayed due to the warm-up phase (during which rewards are not taken into account) but also harmed, presumably because the value estimator has no ability to shape the representation to better suit its needs. The Dyna baseline performs badly while the Dyna* baselines perform relatively well. This is likely due to the delusional transitions generated by the model at the early stages of training, from which the value estimator never recovers. However, the Dyna* baseline does not achieve satisfactory OOD performance (Figure 7), presumably because its planning only focuses on observed data, and hence only improves the in-distribution performance, due to insufficiently strong generalization. The NOSET baseline performs very badly even in-distribution, per Figure 6. In the Appendix, we show that the NOSET baseline seems only able to perform well in a more classical, static RL setting, which may indicate that it relies on memorization. We provide more results regarding the model accuracy in the Appendix.

### 5.3.2 OOD Task-Solving Performance

The OOD evaluation focuses on testing the agents' performance in a set of environments forming a gradient of task difficulty. In Figure 7, we present the performance error bars of

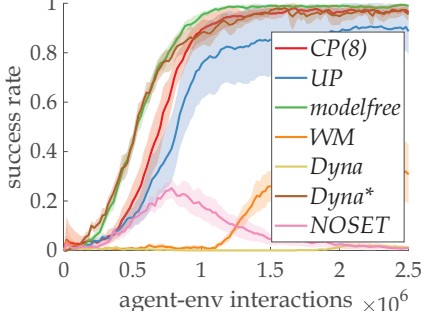

Figure 6: **In-distribution task performance**: the $x$-axis shows the training progress ($2.5 \times 10^6$ agent-environment interactions). The $y$-axis values are generated by agent snapshots at times corresponding to the $x$-axis values. CP, UP, model-free and Dyna* agents all learn to solve the in-distribution tasks quickly. All error bars are obtained from 20 independent runs.

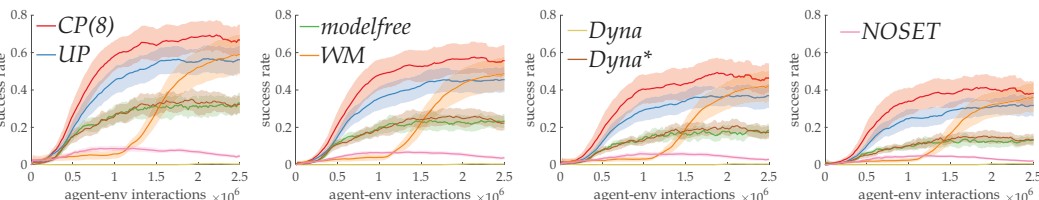

(a) OOD, difficulty 0.25  (b) OOD, difficulty 0.35  (c) OOD, difficulty 0.45  (d) OOD, difficulty 0.55

Figure 7: **OOD performance under a gradient of difficulty.** The figures show a consistent pattern: the MPC-based end-to-end agent equipped with a bottleneck (CP) performs the best. All error bars are obtained from 20 independent runs.

the compared methods under different OOD difficulty levels. CP(8), CP with bottleneck size $n = 8$, shows a clear performance advantage over UP, validating the OOD generalization capability. The Dyna* baseline, essentially the performance upper bound of Dyna-based planning methods, shows no significant performance gain in OOD tests compared to model-free methods. WM may have the potential to reach similar performance as CP, yet it needs to warm up the encoder with a large portion of the agent-environment interaction budget, if no free unsupervised phase is provided. We dive into this matter in the Appendix.

### 5.3.3 Ablation

We validate design choices with ablation. Figure 8 visualizes two of these experiments. For more ablation results, which include validation of the effectiveness of different model choices, and further quantitative measurements, *e.g.* of OOD ability as a function of behavior optimality and model accuracy, please check the Appendix.

### 5.4 Summary of Experimental Results

With the scope limited to our experiments, the results allow us to draw these conclusions:

- Set-based representations enable at least in-distribution generalization across different environment instances in our non-static setting, where the agents are forced to discover dynamics that are preserved across environments;

- Model-free methods seem to face more difficulties in solving our OOD evaluation tasks which preserved the same environment dynamics to the corresponding in-distribution training settings;

- MPC exhibits better performance than Dyna in the tested OOD generalization settings;

- Online joint training of the representation with all the relevant signals could bring benefits to RL, as suggested in [22]. Please check Appendix E for more discussions of this matter;

- In accordance with our intuition, transition models with bottlenecks tend to learn dynamics better in our tests. This is likely for they prioritize learning the relevant aspects, while models without bottleneck may have to waste capacity on irrelevance;

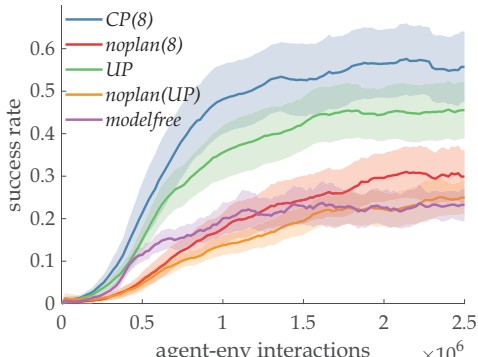

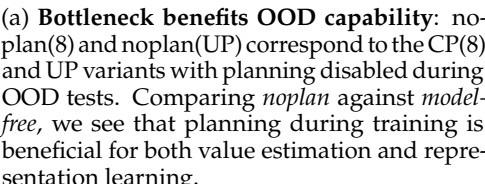

(a) **Bottleneck benefits OOD capability**: no-plan(8) and noplan(UP) correspond to the CP(8) and UP variants with planning disabled during OOD tests. Comparing *noplan* against *model-free*, we see that planning during training is beneficial for both value estimation and representation learning.

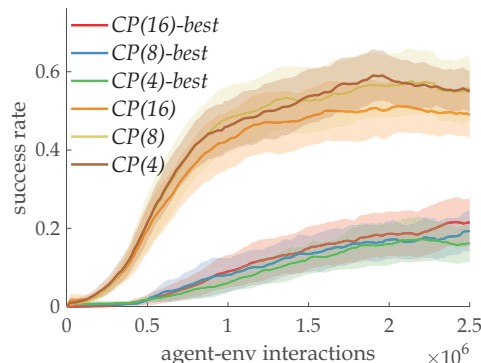

(b) **Value estimators do not generalize well in our OOD tests**: random heuristic significantly outperforms best-first heuristic OOD.

Figure 8: **Key ablation results**: With diff 0.35, each error bar is obtained from 20 independent runs.

- From further experiments provided in the Appendix E, we observe that bottleneck-equipped agents may also be less affected by larger environmental scales, possibly due to their prioritized learning of interesting entities.

## 6   Conclusion & Limitations

We introduced a conscious bottleneck mechanism into MBRL, facilitated by set-based representations, end-to-end learning and tree search MPC. In the non-static RL settings, the bottleneck allows selecting the relevant objects for planning and hence enables significant OOD performance.

One limitation of our work is the experimental focus on only Minigrid environments, due to the need to validate carefully our approach. For future works, we would also like to extend these ideas to temporally extended models, which could simplify the planning task, and are also better suited as a conceptual model of C1. Finally, we note that the architectures we use are involved and can require careful tuning for new types of environments.

## Acknowledgements

Mingde is grateful for the financial support from the Fonds de Recherche du Québec - Nature et Technologies (FRQNT). Yoshua acknowledges the financial support from Samsung Electronics and IBM.

We acknowledge the computational power provided by Compute Canada. We are also thankful for the helpful discussions with Xiru Zhu (about the design of the environment generation procedure), David Yu-Tung Hui (about the bag-of-word representations, insights on BabyAI as well as about the writing of the introduction section), Min Lin (about the design of the dynamics model as well as the early stage brainstorming) and Ian Porada (for consistently supporting the student authors).

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
