# Appendices

## A  Architecture Details

### A.1  Birdseye View of Overall Design

We present the organization of the components for the proposed CP agent in Fig 9. For the model-free baseline agent, we contributed the design of the state set encoder and the set-based value estimator. For the model-based agent, we additionally devised the design of two transition models, one with the conscious bottleneck and another without.

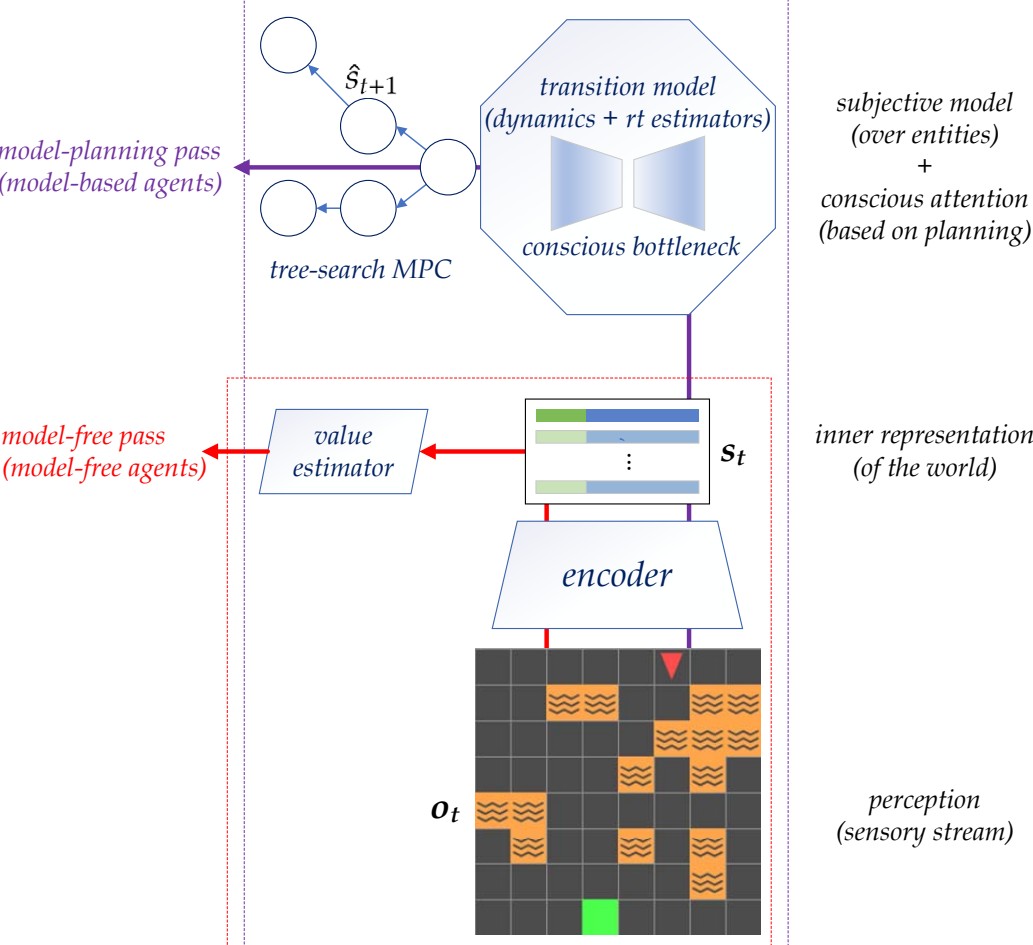

Figure 9: Overall organization of the proposed components for the CP agent. The transition model includes the reward-termination estimator, the dynamics estimator and the optionally the conscious bottleneck. Drawing similarity to the human mind, the 3-layered design corresponds naturally to human perception, inner representation and the conscious planning models.

### A.2  Action-Conditioned Transformer Layer

A classical transformer layer consists of two consecutive sub-layers, the multi-head SA and the fully connected, each containing a residual pass. Similar to the processing of the positional embedding, we first embed the discrete actions into a vector and then concatenate it to every intermediate object output by the SA sub-layer. This way, each transformer layer becomes action-conditioned. An illustration of the component is provided as Figure 10.

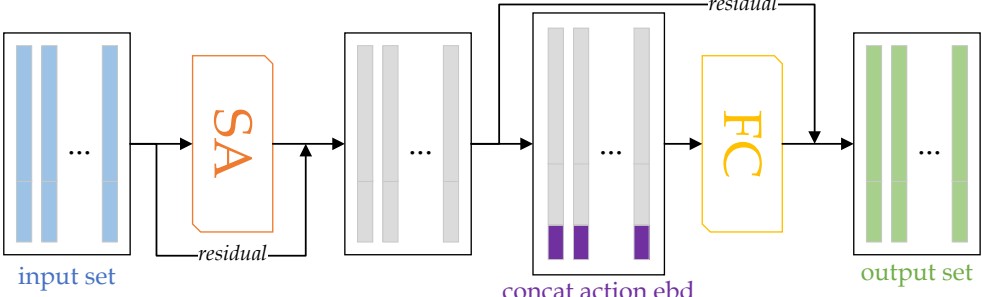

Figure 10: The computational flow of the action-conditioned transformer layer: compared to the classical transformer layers, we concatenate additionally the action embedding to the end of every intermediate object embeddings in the FC pass. The FC pass facilitates $X' = X + f(cat[X, a])$, where $X$ is the set of objects input to the FC part of the action-conditioned transformer layer, $cat([X, a])$ is the concatenation of action embedding $a$ to every object embedding in $X$ and $X'$ is the output set. Note that $f$ downscales the dimensionality of its input to match $X$.

### A.3 Bottleneck Dynamics

The architecture for the bottleneck dynamics (the dynamics operator that simulates $\hat{c}_{t+1}$ from $c_t$, $a_t$) is a stack of action-conditioned transformer layers.

### A.4 Reward-Termination Estimator

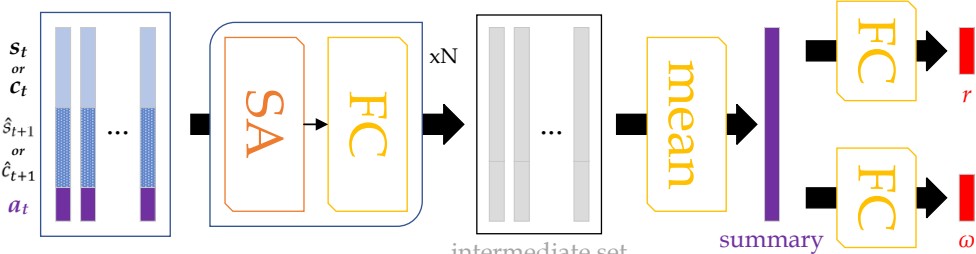

Figure 11: Design of the reward-termination estimator: the state / bottleneck set, the imagined state / bottleneck set as well as the embedding of the action are aligned and concatenated to predict the two outputs. When there is a conscious bottleneck, $c_t$ comes from the selection, $\hat{c}_{t+1}$ is the output of rolling $c_t$ into the dynamics model with $a_t$; When there is not, $\hat{s}_{t+1}$ comes from the forward simulation of the model. With deterministic, it is sufficient to predict the reward and termination with only $s_t$ and $a_t$. This design would be compatible if the dynamics simulation could handle stochastic dynamics.

In the experiments, we wanted functional architectures with minimal sizes for all the components. Thus, globally for the set-input architectures, we have limited the depth of the transformer layers to be $N = 1$ wherever possible. The FC components are MLPs with 1-hidden layer of width 64. Exceptionally, we find that the effectiveness of the value estimator needs to be guaranteed with at least 3-transformer layers. For the distributional output, while the value estimator has an output of 4 atoms, the reward estimator has only 2.

## A.5 Bottleneck Selector

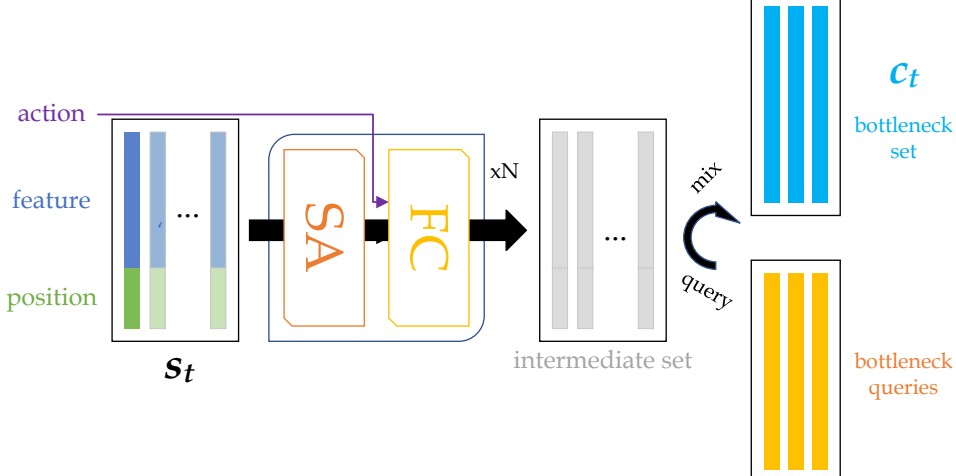

Figure 12: Design of the Bottleneck Compressor: the bottleneck set $c_t$ is obtained by querying the whole set $s_t$ with a learned query set of size $k$, using semi-hard multi-head attention. The selection is conditioned on the chosen action. Please refer to Section B.1 for more details of the query operation.

## A.6 Bottleneck Integrator

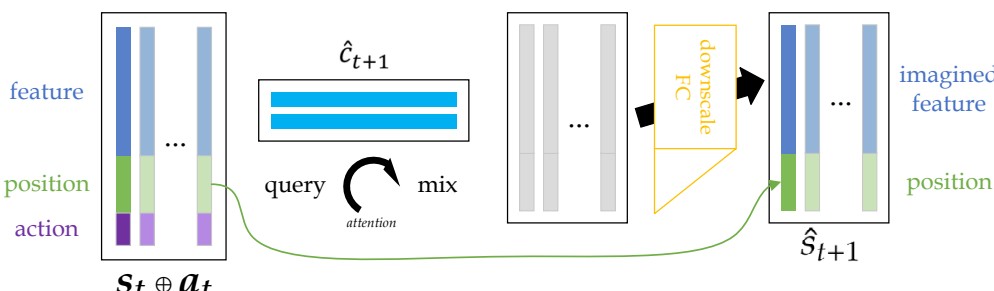

Figure 13: Design of the Bottleneck Integrator: $\hat{s}_{t+1}$ is generated by using the action-augmented $s_t$ to query the imagined bottleneck set $\hat{c}_{t+1}$. Note that there is the similar operation of downscaling objects to features and copying the positional tails. Please refer to Section B.1 for more details of the query operation.

# B Prerequisites

Here, we introduce some prerequisites for better understanding of the used operations.

## B.1 Attention

One of the most important permutation invariant operations on sets of objects is the *attention querying*, which leads to the variants of attention mechanisms [4]. Here, we revisit a generic set query procedure:

For an object to *query* another set of objects, the following steps are taken:

1. The object is transformed into a query vector. This is generally done via linear transformations.

2. The set of objects is independently transformed into two other sets of the same cardinality, named the key set and the value set, respectively.

3. The query vector now compares itself with each key vector in the key set according to some similarity function, *e.g.* scaled dot product, and obtain a vector of un-scaled "attention weights" which is later normalized into a vector with unit $L_1$ norm.

4. The value vectors are weighted by the normalized attention weight vector and combined (typically by linear transformations), yielding the output vector; Querying a set with another set is no different from independently applying the described procedure multiple times. The number of outputs always matches the size of the query set.

Using a set to query itself using the above procedure yield the so-called "self-attention". Using multiple groups of linear transformations and computing the final output from the ensemble of query results is called "multi-head attention". If we erase the lowest attention weights and keep only the top-$k$ ones before the $L_1$ re-normalization, the resulting method is called "semi-hard" attention: for the top-$k$ matches, the attention is soft while for the bad matches, the attention is hard.

## B.2 Distributional Outputs

In this paper, we adopt distributional outputs for the designs of the value and reward estimators. In a nutshell, a distributional output converts a scalar prediction problem with a 1-dimensional output to a predicting a distribution, which is later converted to a scalar by a weighted sum corresponding to the support. This greatly alleviates the problem introduced by the difference in the magnitude of outputs. Please check [6] for more details and [19] for a representative use case.

## C Experiment Insights

**Integer Observations**  For MiniGrid worlds, the observations are consisted of integers encoding the object and the status of the grids. We found that for the UP models with these integer observations, the transformer layers are not sufficiently capable to capture the dynamics. Such problem can be resolved after increasing the depth of the FC layer depth by another hidden layer. This is one of the reasons why we prioritized on using CP models for the observation-level learning of Dyna, *i.e.* CP models can handle integer features without deepening.

Similarly, we have tested the effect of increasing the depth of the linear transformations in SA layers. We did not observe significance in the enhancement of the performance, in terms of model learning or RL performance.

**Addressing Memorization with Noisy Shift**  We discovered a generic trick to enforce better generalization based on our state-set encoding: if we use fixed integer-based positional tails which correspond to the absolute coordinates of the objects, we can add a global noise to all the $x$ and $y$ components in a set whenever one is encoded. By doing so, the coordinate systems would be randomly shifted every time the agent updates itself. Such shifts would render the agent unable to memorize based on absolute positions. This trick could potentially enhance the agents' understanding of the dynamics even if in a classical static RL setting, under which the environments are fixed.

## D Experiment Configurations

The source code for this work is implemented with TensorFlow 2.x and open-source at `https://github.com/PwnerHarry/CP`.

Multi-Processing: we implement a multi-process configuration similar to that of Ape-X [20], where 8 explorers collect and sends batches of 64 training transitions to the central buffer, with which the trainer trains. A pause signal is introduced when the trainer cannot consume

fast enough *s.t.* the uni-process and the multi-process implementation have approximately the same performance, excluding the wall time.

Feature Extractor: We used the Bag-Of-Word (BOW) encoder suggested in [21]. Since the experiments employ a fully-observable setting, we did not use frame stack. In gym-MiniGrid-BabyAI environments, a grid is represented by three integers, and three trainable embeddings are created for the BOW representation. For each object (grid), each integer feature would be first independently transformed into embeddings, which is then mean-pooled to produce the final feature. The three embeddings are learnable and linear (with biases).

Stop criterion: Each runs stops after $2.5 \times 10^6$ agent-environment interactions.

Replay Buffer: We used prioritized replay buffer of size $10^6$, the same as in [19]. We do not use the weights on the model updates, only the TD updates.

Optimization: We have used Adam [27] with learning rate $2.5 \times 10^{-4}$ and epsilon $1.5 \times 10^{-4}$. The learning rate is the same as in [30]. Our tests show that using $6.25 \times 10^{-5}$, as suggested in [19], would be too slow. The batch size is the same for both value estimator training and model training, 64. The training frequency is the same as in [19]: every 4 agent-environment interactions.

$\gamma$: Same as in [19]. 0.99.

Transformers: For the SA sublayers, we have used 8 heads globally. For the FC sublayers, we have used 2-layer MLP with 64 hidden units globally. All the transformer related components have only 1 transformer layer except for that of the value estimator, which has 3 transformer layers before the pooling. We found that the shallower value estimators exhibit unstable training behaviors when used in the non-static settings.

Set Representation: The length of an object in the state set has length 32, where the feature is of length 24 and the positional embedding has length 8. Note that the length of objects must be dividable by the number of heads in the attentions. The positional embeddings are trainable however their initial values are constructed by the absolute $xy$ coordinates from each corner of the gridworld ($4 \times 2 = 8$). We found that without such initialization the positional embedding would collapse.

Action Embedding: Actions are embedded as one-hot vectors with length 8.

Planning steps: for each planning session, the maximum number of simulations based on the learned transition model is 5.

Exploration: $\epsilon$ takes value from a linear schedule that decreases from 0.95 to 0.01 in the course of $10^6$ agent-environment interactions, same as in [19]. For evaluation, $\epsilon$ is fixed to be $10^{-3}$.

Distributional Outputs: We have used distributional outputs [6] for the reward and value estimators. 2 atoms for reward estimation (mapping the interval of $[0, 1]$) and 4 atoms for value estimation (mapping the interval of $[0, 1]$).

Regularization: We find that layer norm is crucial to guarantee the reproducibity of the performance with set-representations. We apply layer normalization [1] in the sub-layers of transformers as well as at the end of the encoder and model dynamics outputs. This applies for the NOSET baseline as well.

Modelfree baseline: We did not use the full Rainbow agent [19] as the baseline for that we want to keep our agent as minimalist as possible. The agent does not need the dueling head and the noisy net components to perform well, according to our preliminary ablation tests.

# E  More Experimental Analyses

## E.1  In-Distribution Model Accuracy

We intend to demonstrate how well the bottleneck set captures the underlying dynamics of the environments. For each transition, we split the grid points into two partitions: one containing all relevant objects that changed during the transition or have an impact on reward or termination, while the other contains the remaining grid points. As a result, the dynamics error is split into into two terms which correspond to the accuracy of the model simulating the relevant and irrelevant objects respectively.

Acknowledging the differences in the norm of the learned latent representations, we use the element-wise mean of $L_1$ (absolute value) difference between $\hat{s}_{t+1}$ and $s_{t+1}$ but normalize this distance by the element-wise mean $L_1$ norm of $s_{t+1}$, as a metric of model accuracy, which we name the *relative L1*. This metric shows the degree of deviation in dynamics learning: the lower it is, the more consistent are the learned and observed dynamics.

Figure 14 (a) presents the *relative L1* error of the a CP configuration during the in-distribution learning. With the help of the bottleneck, the error for the irrelevant parts converge very quickly while the model focuses on learning the relevant changes in the dynamics. Additionally, we provide the model accuracy curves of the WM and Dyna baselines in the Appendix.

For reward and termination estimations, our results show no significant difference in estimation accuracy with different bottleneck sizes. However, they do seem to have significant impact on the dynamics learning. In Figure 14 (b), we present the convergence of the relative dynamics accuracy of different CP and UP agents. CP agents learn as fast as UP, which indicates low overhead for learning the selection and integration.

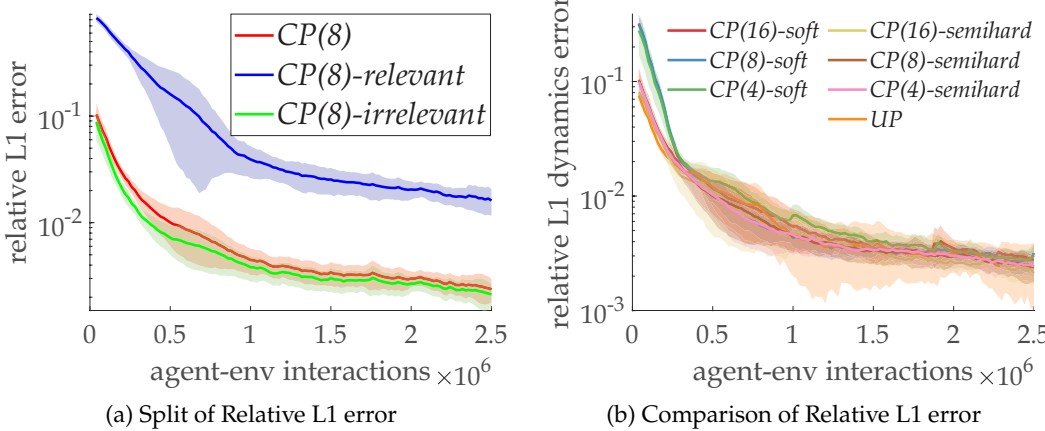

(a) Split of Relative L1 error          (b) Comparison of Relative L1 error

Figure 14: Curves showing in-distribution evaluation: Each band shows the mean curve (bold) and the standard deviation interval (shaded) obtained from 20 independent seed runs. a) Partitioning of the relative L1 dynamics prediction errors into that of the relevant objects and the irrelevants: The difference in the errors shows that the bottleneck learns to ignore the irrelevance while prioritizing on the relevant parts of the state; b) Comparison of the overall relative L1 errors (not partitioned). For CP variants, the numbers in the parentheses correspond to the bottleneck sizes and the suffixes the types of attention for the bottleneck selection. Semi-hard attention learns more quickly than soft attention at early stages but they both converge to similar accuracy levels. This is likely due to the fact that semi-hard attention is forced to pick few objects and thus to ignore irrelevant objects even at early stages of training.

## E.2  More Ablation Results

Figure 15 visualizes more experiments which highlight the effectiveness of the bottleneck's contribution towards OOD generalization.

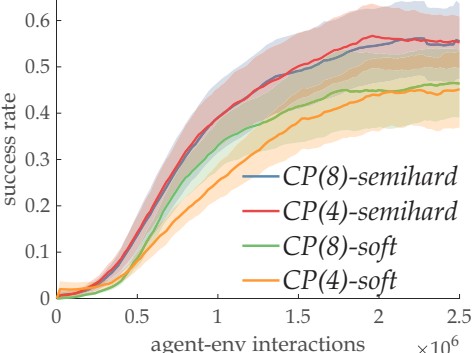

(a) Attention Type: semi-hard attention outperforms better when used in bottleneck selection

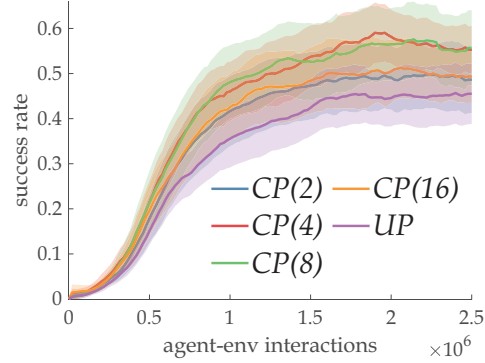

(b) Bottleneck Size: bottleneck sizes 4 and 8 perform similarly the best within $\{2, 4, 8, 16\}$. Also, the performance with bottlenecks is consistently better than that without (UP), showing the bottlenecks' effectiveness for OOD generalization

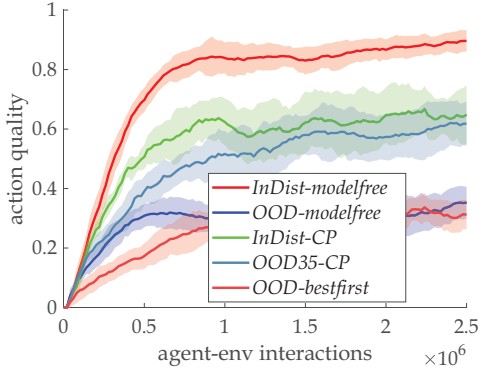

(c) Action Quality: we record if the actions taken by the methods are optimal. For in-distribution evaluation, the methods both perform well. Interestingly, the model-free agent performs superior possibly due to its simple value-based greedy policy. However in OOD evaluation, only the CP agent with the random heuristic shows neither significant deterioration nor signs of overfit in the action qualities.

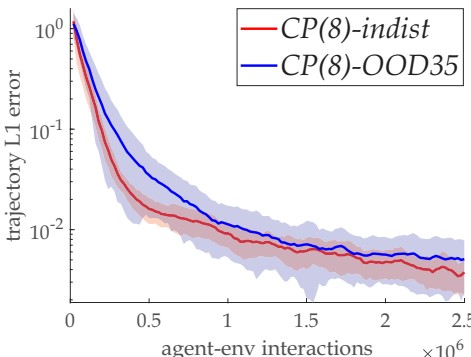

(d) Tree Search Dynamics Accuracy: the curves show the cumulative L1 error of the chosen trajectory during tree search. These are obtained by comparing the imagined states simulated through multi-step planning with the help of a perfect environment model. The curves show no signs of overfit as the cumulative trajectorial dynamics accuracy during OOD evaluation is growing over time.

Figure 15: Ablation results with difficulty 0.35: each band is consisted of the mean curve and the standard deviation interval shades obtained from 20 independent seed runs.

### E.3 Planning Steps

Intuitively we know there should be a good value for the planning step hyperparameter. If the planning steps are too few, then the planning would have little gain over model-free methods. While if the planning steps are too many, we suffer from cumulative planning errors and potentially prohibitive wall time. We tried different number of planning steps for 8-picks semi-hard CP. Note that the planning steps during training and OOD evaluation are equivalent. Such particular choice is to make sure that the planning during evaluation would be carried out to the same extent during training. The results visualized in 16 suggested that 5 planning steps achieves the best performance in OOD with difficulty 0.35.

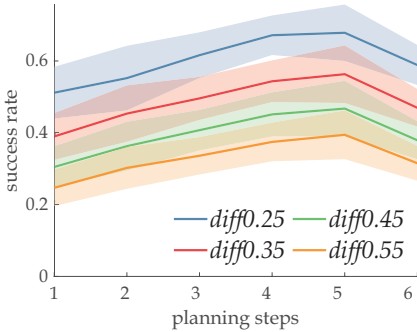

Figure 16: Success rate of CP(8) agent under OOD difficulty 0.35. Note that for each agent variant, the planning steps used in training and OOD evaluation are the same.

### E.4 Action Regularization

We applied an additional regulatory loss that predicts the action $a_t$ with $c_t$ and $\hat{c}_{t+1}$ as input, resembling the essence of an inverse model [12]. The loss is a unscaled categorical cross-entropy, like that of the termination prediction. This additional signal is shown in experiments to produce better OOD results, especially when the bottleneck is small, as visualized in Figure 17.

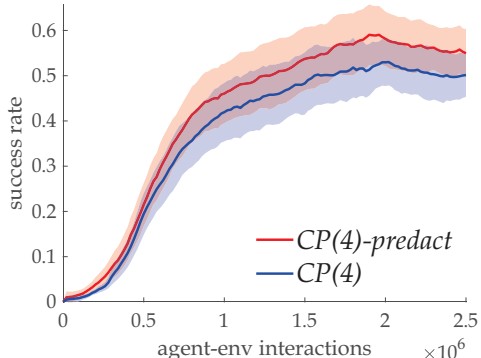

Figure 17: Impact on the success rate of CP(4) agents under OOD evaluation with difficulty 0.35 by the action regularization loss in the bottleneck. The "predact" configuration is by default enabled in the main manuscript, *i.e.* all the CP results shown except in this figure has action regularization enabled. Each point of the band correspond to the mean and standard deviation of the success rate of OOD evaluation during the last $5 \times 10^5$M agent-environment interactions (last 20% training stage).

### E.5 Potential of WM Baseline

In case the readers are curious about how the WM baseline would evolve after the $2.5 \times 10^6$ steps cutoff, we provide an additional set of experiments featuring a free unsupervised

learning phase of $10^6$ agent-environment interactions. As illustrated in Figure 18, observations suggest that WM baseline could not achieve similar performance as that of CP due to that the representation is not jointly shaped for value estimation. The results show promise of the methodology of representation learning with joint signals. However, this is not to say that an unsupervised learning of a world model is not beneficial in general, just limited to this case and this planning methodology.

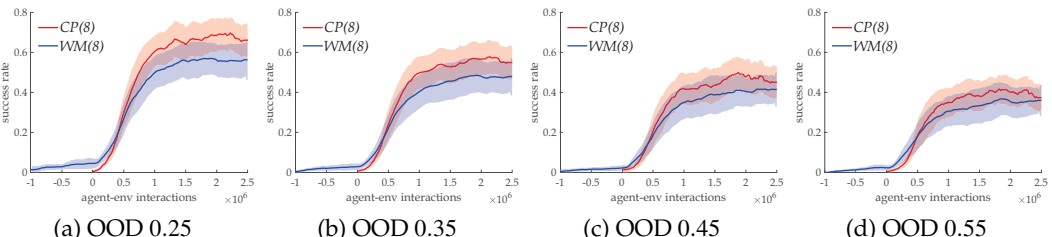

| (a) OOD 0.25 | (b) OOD 0.35 | (c) OOD 0.45 | (d) OOD 0.55 |

Figure 18: **OOD performance comparing CP and WM under a spectrum of difficulty.** WM(8) is the WM baseline which uses the same architecture as CP(8), for fair comparison. The WM(8) results are shifted for a free unsupervised world model learning phase of $10^6$ steps. All error bars are obtained from 20 independent runs.

## E.6 Tests on Different World Sizes

To inspect the scalability of the proposed method, we compare the methods CP(8), UP and model-free in a gradient of gridworld sizes. The results are presented in Figure 19.

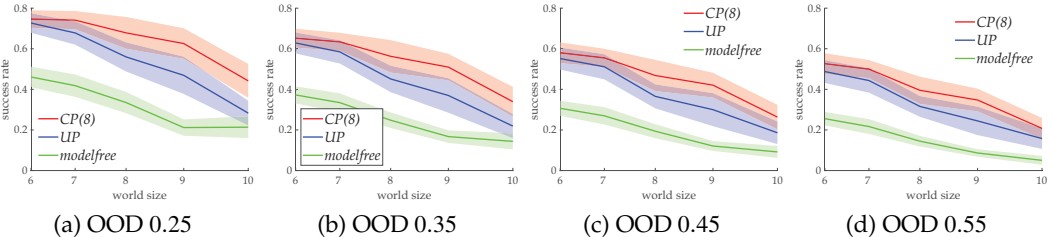

| (a) OOD 0.25 | (b) OOD 0.35 | (c) OOD 0.45 | (d) OOD 0.55 |

Figure 19: **OOD performance under a spectrum of difficulty and world sizes.** The *x*-axes are ticked with #grids in each gridworld size, representing the number of entities for in the state set, thus non-uniform. The smaller the world sizes, the better and the closer the performance of the three methods are. The fact that the CP(8) performance deteriorates slower than UP suggests that the bottleneck may contribute to more scalable performance in tasks with larger amount of entities. All error bars are obtained from 20 independent runs.

## E.7 Different Tasks

To test the applicability of CP on more scenarios, we craft some additional sets of experiments with MiniGrid to test the robustness. For these extra sets of experiments, we prioritize on presenting the comparison of the CP, UP and modelfree agents' performance.

## E.8 Alternative Dynamics

First, we want to see if the experimental conclusions would still hold on a task with different action dynamics. Thus, we modify the original task in the main manuscript by a new set of Turn-And-Forward dynamics: the action space is re-resigned to include 4 composite actions which first turns to some directions (forward, left, right or back based on the current facing direction) and then move forward if possible (if not stepping out of the world).

Intuitively, this set of new dynamics can be seen as a composition of the original and hence is easier to solve and more effective in terms of planning, *i.e.* the same number of planning

steps would lead deeper into the future. In Figure 20, we observe that all three methods are performing better compared to the original tasks and the experimental conclusions are re-validated.

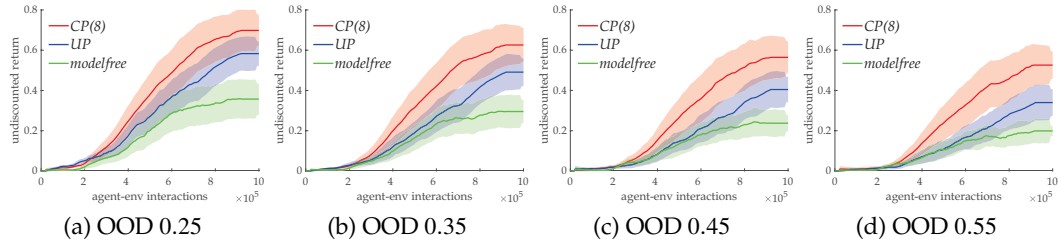

(a) OOD 0.25    (b) OOD 0.35    (c) OOD 0.45    (d) OOD 0.55

Figure 20: **OOD performance under a gradient of difficulty in Turn-and-Forward tasks.** All error bars are obtained from 20 independent runs.

### E.8.1 Cluttering Effect

For the second set, upon the turn-and-forward environments, we add randomly changing colors to every grid so that a cluttering effect is posed to hinder the agents from understanding the object interactions as well as disturbing the bottleneck selection. Specifically, the distracting colors are sampled uniformly randomly from 6 possibilities for each grid of each observation. We add one additional baseline named CP(8)+, which denotes a CP(8) agent with noise injection at the input of the dynamics model. Specifically, we sample an 8-dimensional **0**-mean identify variance Gaussian noise that is replicated and concatenated to every bottleneck object.

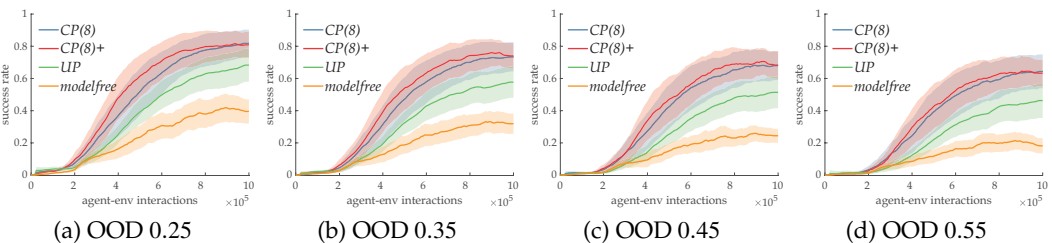

(a) OOD 0.25    (b) OOD 0.35    (c) OOD 0.45    (d) OOD 0.55

Figure 21: **OOD performance under a gradient of difficulty in Turn-and-Forward tasks with color distractions.** All error bars are obtained from 20 independent runs.

From Figure 21, we observe that CP(8) still performs better than UP therefore re-validating the effectiveness of the bottleneck mechanism. It is likely that our state set encoder to learn to ignore the distractions and thus make the bottleneck selector to be able to direct attention to the relevant objects as what we have done for the tasks with the old dynamics. Furthermore, CP(8)+ seems to achieve similar performance as CP(8) but converges faster. This experimental observation suggests that noisy inputs may be beneficial to the learning behavior of the dynamics model for a noisy environment. Yet across the paper, we have tried not to use any other additional means to enhance the performance of our agents in order to isolate impact of unwanted components.

### E.8.2 Key-Chest Unlocking

We built the final set of experiments built upon the logic of MiniGrid's MiniGrid-Unlock-v0 and Turn-and-Forward dynamics. The agent needs to navigate the gridworld while avoiding obstacles (0 reward, end of episode) to get a key first (+0.5 reward) and then unlock a chest (+0.5 reward, end of episode, considered as success) to finish the task.

In this task in particular, the bottleneck is tested against a conditional selection: the reward and termination feedbacks on the chest is conditioned on if the key is still not acquired.

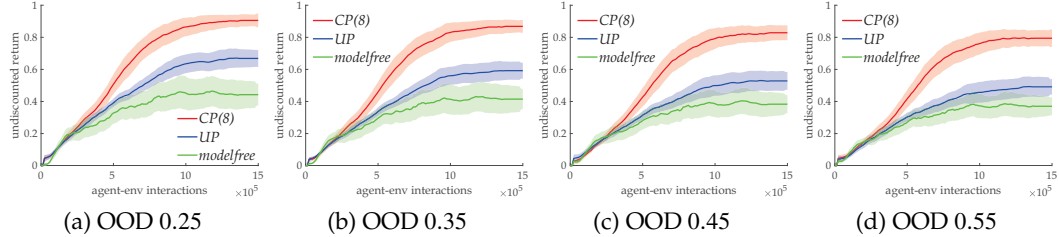

| (a) OOD 0.25 | (b) OOD 0.35 | (c) OOD 0.45 | (d) OOD 0.55 |

Figure 22: **OOD performance under a gradient of difficulty in Key-Chest Unlock Task.** The $y$-axes values are undiscounted cumulative episodic return. All error bars are obtained from 20 independent runs.

Hence, in the bottleneck, when the agent plans to step into the chest, the bottleneck should additionally select the key to predict well the outcome. From Figure 22, the results of the three compared methods are more differentiable than the original turn-or-forward setting and the pattern remains consistent across all settings, the bottleneck-equipped CP agent performs better than UP and modelfree.

### E.9 Learning Capable Representation with Non-Conflicting Joint Signals

We have gathered more empirical evidence regarding the non-conflicting training of the state-representation based on the signals. In terms of model learning accuracy, according to our results related to WM baseline, removing the value estimation signal would result in poorer representation but not lower accuracy when predicting other relevant signals; Removing termination signal would not impact the convergence of reward prediction accuracy or that of the state prediction however the RL performance is hindered. Removing the reward signal or the next state prediction signal leads to total collapse of the tree-search based behavior policy however the convergence of the remaining model training signals is not affected much. With these, we would like to suggest that we have, at least in this task setting, learned a set-based representation capable of predicting all interesting quantities.

## F   Visualization of Selection

We present some visualization of the object selection during the planning steps in Figure 23. In (a), with the intention of turning left, the agent takes into the bottleneck the location of itself within the grid (visualized as the teal triangle with white surroundings, color-inverted from red-black); For (b), the agent additionally pays attention to the lava grid on its right while trying to turn right. In (a) and (b), the goal square (pink, color-inverted from green) is also paid attention but we cannot interpret such behavior. Finally in (c), we can see that the agent takes consideration into the grid (the blue lava grid, color-inverted from orange) that it is facing before taking a step-forward action. Though these visualization provides an intuitive understanding to the agents' behavior, they do not serve statistical purposes.

We additionally have collected the coverage ratio of all the relevant objects by the selection phase in all the in-distribution and OOD evaluation cases along the process of learning. The collected data on bottleneck sizes 4, 8 and 16 indicate that the coverage is almost perfect very early on during training. We do not provide these curves because the convergence to 100% is so fast that the curves would all coincide with line $y = 1$, with some minor fluctuations of the standard deviation shades.

## G   Details of Compared Baseline Methods

### G.1   Staged Training (World Models)

The agents with World Model (WM) trained in stages share the same architectures as their CP or UP counterparts. The main difference is that the WM agents adopt a 2-staged training

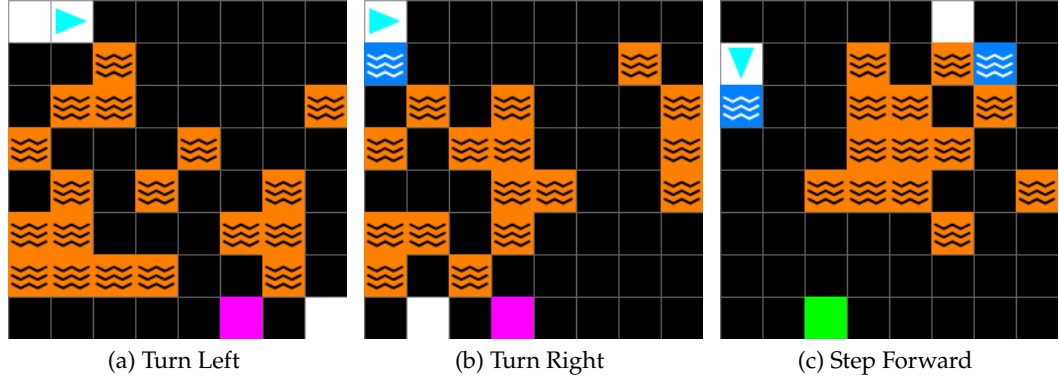

| (a) Turn Left | (b) Turn Right | (c) Step Forward |

Figure 23: Visualization of the bottleneck selection given the observation and specific actions. These figures are extracted from a fully trained CP(2) agent under OOD evaluation. The bottleneck is set to very small for clearer visualization purposes. We invert the color of the selected objects by the best performing head, *i.e.* the head that covers the most relevant objects, though the selection quality would be sufficiently justified if *all* the heads could *cumulatively* cover all the interested objects. The grids of the selection would be at least 2 but at most 4 due to the design.

strategy: In the first $10^6$ agent-environment interactions, only the model is trained and therefore the representation is only shaped by the model learning. In the first stage the agent relies on a uniformly random policy. After $10^6$ interactions, the agent freezes its encoder as well as the model to carry out value estimator learning. Note that the agent carries out tree-search MPC with the frozen model in the second stage. Compared to CP or UP, the exploration scheme is delayed but unchanged. Also, the training configurations do not change.

### G.2  Dyna

The Dyna agents share the model-free part of the architecture as CP or UP. The models that Dyna baselines learn are powered by our action-conditioned set-to-set architecture on an observation-level. The training timings for both the model and the value estimator are not changed, though they do not jointly shape the representation and are used very differently compared to CP or UP. In our implementation, the generation of imagined transitions is carried out by dedicated processes. These processes generate transitions and send them to a dedicated global replay buffer of size 1024. The small size is to ensure that the delusional transitions would be washed out soon after the model is effective. The TD learning of the value estimator samples a double-sized mini-batch, half from the buffer of real transitions and half imagined. While the model training uses only the true transitions, with unit-sized batches. Since our model is not generative, we rely on free-of-budget model-free agents to collect true $\langle s, a \rangle$ pairs from the environment and then complete the missing parts of the transitions (reward, termination and next observation) using the model (for the Dyna baseline with true dynamics, we just collect the whole transition exclude the model). This way, the transitions would follow the state-action occupancy jointly defined by the MDP dynamics and the policy. The approach is a compromise to implement a correctly performing Dyna agent with a non-generative model.

### G.3  NOSET

The NOSET baselines embraces traditional vectorized representations. We use the same encoder but instead of transforming the feature map into a set, we flatten it and the linearly project it to some specific dimensionality (256). This vector would be treated as $s_t$, the same as the most existing DRL practices. Since all set-based operations would be now obsolete for the vectorized representation, they are substituted with 3-layered FCs with hidden width 512. The 2-layered dynamics model employ a residual connection with the expectation that the model might learn incremental changes in the dynamics. In our experiments with

randomly generated environments for each episode, the NOSET baseline performs miserably. However, if we instead randomly generate an environment at the start of the run and use the same one for the whole run, *i.e.* adopt the more classical RL setting, we find that the NOSET baseline is able to perform effectively, as shown in Figure 24.

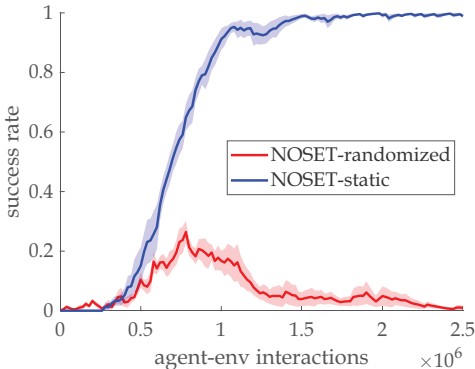

Figure 24: NOSET baseline performance on randomized and static random environments. Each band is consisted of the mean curve and the standard deviation interval shades obtained from 20 independent seed runs.

The dimension of the state representation, the widths and the depths of the FC layers are obtained through coarse grid tuning of the exponents of 2. We find that architectures exceeding the chosen size are hardly superior in terms of performance.

## H   Tree Search MPC

The agent (re-)plans at every timestep using the learned model in the hidden state level. The in-distribution planning strategy is a best-first search MPC heuristic. While the OOD planning heuristic is random search. Note that no matter which heuristic is used, the chosen action is always backtracked by the trajectory with the most return.

We present the pseudocode of the tree search MPC in Algorithm 1. Additionally, we provide an example showing how the best-first heuristic works in an assumed decision time with $\gamma = 1$ and $|\mathcal{A}| = 3$ and maximum planning steps 3.

## I   Failed Experiments

We list here some of our failed trials along our way of exploring the topic of this work.

### I.1   Straight-Through Hard Subset Selection with Gumbel

We initially tried to use Gumbel subset selection [49] to implement a hard selection based bottleneck but to no avail. We expect the model to pick the right objects by generating a binary mask and then use the masked objects as the bottleneck set. This two-staged design would align more with the consciousness theories and would yield clearer interpretability. However, it suffers from an implicit chicken-and-egg problem that we have not successfully addressed: to learn how to pick, the model should first understand the dynamics. Yet if the model does not pick the right objects frequently enough, the dynamics would never be understood. Our proposed semihard / soft approaches address such problem by essentially making the two staged selection and simulation as a whole for the gradient-based optimization.

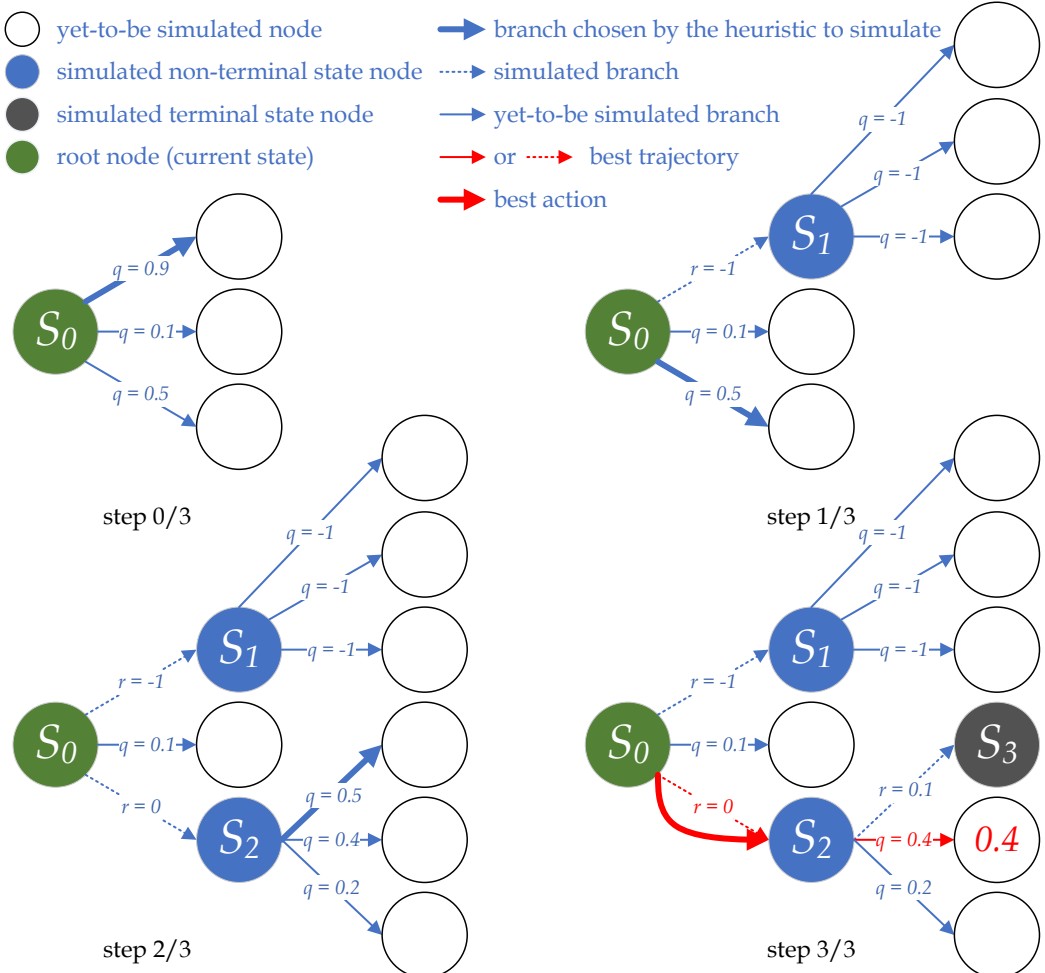

Figure 25: Example of the Best-First Heuristic: Step 0 / 3) Start of planning, with the root node and three branches. The branch $\langle s_0, a_0 \rangle$ is chosen due to the best-first heuristic. If we employ the random search heuristic, like what we do in OOD evaluation, a random branch would be chosen; Step 1 / 3) We expand the chosen branch, popped out of the priority queue. A new node is constructed, together with its out-reaching branches, which are added to the queue. Now the queue has 5 branches in it. The heuristic marks $\langle s_0, a_2 \rangle$ to be the next simulated branch; Step 2 / 3) Simulation of $\langle s_0, a_2 \rangle$ is finished and $\langle s_2, a_0 \rangle$ is marked; Step 3 / 3) Node S3 is imagined via $\langle s_2, a_0 \rangle$ but it is estimated to be a terminal state. Now, the tree search budget is depleted. We locate the root node branch $\langle s_0, a_2 \rangle$ which leads to the trajectory with the most promising return 0.4.

---
**Algorithm 1:** Prioritized Tree-Search MPC
---
**Input:** $s_0$ (current state), $\mathcal{A}$ (action set), $\mathcal{M}$ (model), $Q$ (value estimator), $\gamma$ (discount)
**Output:** $a^*$ (action to be taken)
$q$ = queue(); $q_T$ = queue() //$q_T$ for terminal nodes
$n_u$ = NODE($s_0$, root = True) //$n_u$ denotes a node with branches unprocessed nor in $q$
**while** *True* **do**
    **if** $n_u.\omega$ **then**
        $q_T$.add($\langle n_u, n_u.\sigma \rangle$) //identified as a terminal state. $n_u$ is added to $q_T$ using
        bisection, together with the discounted sum of the simulated rewards along the
        way $n_u.\sigma$
    **else**
        **for** $a \in \mathcal{A}$ **do** $q$.add($\langle n_u, a, n_u.\sigma + \gamma^{n_u.\text{depth}} \cdot Q(n_u.s, a) \rangle$) //bisect *w.r.t.* priority ;
    **if** *isempty*($q$) **then break** //tree depleted;
    $n_c, a_c, v_e$ = $q$.pop() //get branch with highest priority; for in-distribution setting,
     priority is the estimated value of the leaf trajectory
    **if** *budget depleted* **then break** //termination criterion met;
    $\hat{s}, \hat{r}, \hat{\omega} = \mathcal{M}(n_c.s, a_c)$ //simulate the chosen branch
    $n_u$ = NODE($\hat{s}$, parent = $n_c$)
    **if** $n_c.depth > 0$ **then** $n_u.a_b = n_c.a_b$; **else** $n_u.a_b = a_c$ //descendants trace root action;
    $n_u.\omega = \hat{\omega}$; $n_u.\sigma = n_c.\sigma + \gamma^{n_c.\text{depth}} \cdot \hat{r}$
$n_c, a_c, v_e$ = $q$.pop('highest value') //get branch with highest **value** within the
 expandables
$n^* = n_c$;
**if** $\neg$*isempty*($q_T$) **then**
    $n_T = q_T$.pop('highest value') //get node with highest **value** within simulated
     terminal states
    **if** $n_T.value \geq v_e \vee$ *isempty*($q$) **then** $n^* = n_T$;
**if** *isroot*($n^*$) **then** $a^* = a_c$; **else** $a^* = n^*.a_b$;

---

## J More Discussions on Limitations & Future Directions

This paper serves as a proof-of-concept of an interesting research direction: System-2 DRL. It is healthy to point out the limitations of this work as well as some interesting future research directions:

- This paper does not solve the "planning horizon dilemma", a fundamental issue of error accumulation of tree search expansion using imperfect models [23]. We strongly believe that incorporating temporal abstraction of actions, *e.g.* options or subjective time models [51] would gracefully address such problem. Promising as this is, introducing temporal abstraction to model-based RL is non-trivial and requires considerate investigation.

- Constant replanning may be prohibitive in reaction-demanding environments, especially when equipped with a computationally expensive set-based transition model. A planning strategy could be devised to control when or where for the agent to carry out planning, through the means of capturing uncertainty.

- The CP model cannot yet learn stochastic dynamics. The difficulty lies in the design of a compatible end-to-end trainable set-to-set machinery. We would like to address this in future.

## K Potential Negative Societal Impacts

We do not anticipate potential negative societal impacts since this paper is fundamental research regarding reinforcement learning methodology.