# OpenReview forum: "A Consciousness-Inspired Planning Agent for Model-Based Reinforcement Learning"
_NeurIPS.cc/2021/Conference — NeurIPS 2021 Poster_

### Official Review · Reviewer_hG23 · 2021-07-11

**Rating:** 7
**Confidence:** 3

**Summary:**

The authors take inspiration from theories of human consciousness to construct a model-based architecture for RL, with the goal of generalizing more effectively.  They conduct experiments on a gridworld to show that their proposed architecture generalizes more effectively.

**Limitations And Societal Impact:**

Yes.

**Main Review:**

After reading the authors' responses, the new results, and the other reviews, I believe that my concerns have been mostly addressed.  I am raising my score from a 5 to a 7.  (I am tempted to give an 8 or 9, as this paper has some fascinating and exceptionally well-motivated ideas. However, I cannot give an extremely high score to an empirical-only paper which tests only on gridworlds, despite the fact that there are good reasons why more complex environments are beyond the scope of this work and the fact that the authors did add a new gridworld variation.)  I hope the authors follow up this work by showing how to apply similar ideas to more complex environments.

-----------------------

Sections 1-4 (relatively small concerns):

The encoder description in Section 3 (“We use the features...dynamics model, discussed below”) was insufficient and vague; I read it several times, but have no real idea how the encoder works.  A more precise definition of the encoder would strengthen this work (in the appendix if there’s no room in the main paper). Edit: while re-reading, I noticed that Figure 1 is describing this.  I think simply adding a reference to Figure 1 in the text (I don’t think there is any existing reference to Figure 1) might be sufficient to address most of this concern.  However, I’m still confused about how “This approach is different from the practice of adding positional information onto the features”, so clarifying some more in the appendix might still be helpful.

TD loss: “In experiments, a distributional output is used for both value and reward estimation, making this loss a KL-divergence.” This is a little vague and confusing; I’d suggest defining it more precisely.

Section 5 (significant concerns):

-Low number of trials (5 runs): This is too low to make the claims that are made in Section 5.  RL algorithms are notoriously unstable between trials, and the differences shown could easily not appear in the true curves (that is, given an infinite number of trials).  I suggest 1) running MANY more trials, and plotting the standard error instead of standard deviation, to show that the difference is significant, and/or better yet, 2) do proper statistical hypothesis testing to show your conclusions are valid given the data.  For either approach, you will certainly need many more than 5 trials.  This concern is somewhat mitigated in 5.4.1 by having effectively 20 trials.  Still, it would be nice to see the 4 5-trial plots combined as a single plot with 20 trials and standard error bars, and/or see some statistical hypothesis testing (and far more than 20 trials would be better).

-Gridworld: for an entirely empirical paper, only running gridworld experiments is not ideal.  The claim is “CP allows better generalization”, but at best (ignoring the concern above) what is shown is “CP allows for better generalization on 8x8 Gridworlds”, which is not a huge contribution.  More environments (including more difficult/complex ones) would help alleviate this problem.

Minor edits:

-Figure 1 caption, add an article adjective before CNN: “e.g., a CNN”

-Too informal: “Luckily, our feature-position...”

Strengths:

-The plots of different bottleneck sizes are a great idea.  However, I’m not 100% satisfied; more sizes and more trials (runs) per size would be nice.

-Good limitations and failed experiments sections in the appendix.

-Many of the hyperparameter selections are justified in the appendix via references.  This is great and more papers should do this (otherwise the conclusions drawn may be based on the effects of the hyperparameter search results rather than the effects of the differences between the algorithm and baseline).

Question/suggestion:
Are the losses weighted when summed to produce the total loss?  If so, this information should be included (that is to say, the total loss equation is wrong).  If not, this is surprising; I’d suggest giving some intuition about why these losses are simply summed without any weighting and why that works.  Perhaps some empirical evidence in the ablation showing that the summed losses all tend to be a similar order of magnitude, and all help achieve good performance.  In other words, if you remove any of the losses, do the results get significantly worse?  Or are there one or two losses that are not affecting much?

Summary:

While the ideas the authors propose are fascinating and well-motivated, this paper has no theoretical contribution (which is fine for a strong empirical paper), and a somewhat weak empirical contribution.  Nonetheless, it is an interesting paper and has its strengths.  I recommend continuing to refine this work by strengthening the experimental results, as discussed in more detail above.


**Time Spent Reviewing:**

~4.5

---

> ### Author Response · Authors · 2021-08-10
> **A Point-By-Point Reply to Your Comments**
>
> Thank you very much for the helpful comments and the detailed inspection of this work! For important updated figures please check this anonymous link: https://bit.ly/3ivcbEC.
>
> We would like to address your concerns point-by-point.
>
> **Relevant Small Concerns:**
>
> The reference to the encoder mechanisms would be added (from the text to the figure). We have also improved our wording a bit on this matter. While we concatenate a positional embedding onto a feature embedding to create an object embedding, most of the existing work would ADD a positional embedding of the same shape to the feature embedding, mixing them two in a way that triggers the alignment problem (as discussed in the manuscript). We added a figure and a paragraph of explanation into the Appendix so that it would not cause more confusion.
>
> The details of the distributional outputs are expanded more thoroughly in the appendix, with citation links to its origins and popular usages.
> The minor edits are also applied according to your suggestions.
>
> **Major Concerns:**
>
> On low number of trials: we had not been able to do so before the submission for the lack of computational power. we have expanded all seed runs to 20 (included all the new ones added) and correspondingly updated all results. Particularly, more trials regarding the bottleneck sizes are available for your interest, as per Figure 15 (b). Despite small changes in the details of the curves, the conclusions remain the same (however moderated as suggested by reviewers). Results of hypothesis testing for methods in Figure 7 and 8(a) has been added to the Appendix (using the smoothed performance of the last 0.1M steps). This solution is because that we want to preserve the std-based error bars in the figures in accordance to the community norm, yet also wanting to provide the confidence intervals in a separate table for the testing.
>
> **We acknowledge the limitation of relatively simple experimental settings and seek to address it with the following two improvements:**
>
> 1.	Extended results to different sizes of gridworlds of the original set of experiments (with both turn-and-forward and turn-or-forward action spaces). We provide additional results on from 6x6 to 10x10 world sizes. The conclusions regarding the bottleneck are consistent. Based on our observations, world sizes smaller than 6x6 would pose too simple a challenge for differentiated performance (and chances for memorization) while starting from 10x10 the difficulty of exploration kicks in (the variance in the curves becomes large) and the computational requirements become very harsh.
>
> 2.	A newly crafted family of Key-Door tasks is built upon the logic of MiniGrid’s Unlock-v0 and our previous task (with turn-and-forward dynamics). The agent needs to navigate the gridworld while avoiding obstacles (0 reward, end of episode) to get a key first (+0.5 reward) and then unlock a chest (+0.5 reward, end of episode, considered as success) to finish the task. Besides all the other examinations we have done for the previous settings, in this task in particular, the bottleneck is tested against a conditional selection: the reward and termination feedbacks on the chest is conditioned on if the key is still not acquired. Hence, in the bottleneck, when the agent plans to step into the chest, the bottleneck should additionally select the key to predict well the outcome. We prioritize on investigating such effect with the accuracy regarding the selection. The results of the three compared methods are more differentiated than the original turn-or-forward setting and the pattern remains consistent across all settings, the bottleneck-equipped CP agent performs better than UP and modelfree.
>
> I hope that these new results would improve your confidence on the applicability of the method while we explore other possible testing settings which would serve our purpose. You may want to check parts of our rebuttal to Reviewer sQ3E (in order 1 from our view) on the difficulty of finding or crafting suitable experimental settings instead of using Atari, etc.
>
> For your final question regarding the loss aggregation: yes, the losses are added without re-weighting. We think that there could be two factors contributing to the easiness. First, except for the prediction error of the imagined next state, all other signals are distributional regression losses approximately in the same magnitude such as cross entropy of termination prediction and KL of reward and value estimations. Second, apart from regularizing the state features, layernorm helped the losses take value approximately in the same order of magnitude as are the state features.
>
> We have gathered more ablation for your interested empirical evidence regarding the non-conflicting training of the state-representation based on the signals. In terms of model learning accuracy, according to our results related to WM baseline, removing the value estimation signal results in poorer representation but not lower accuracy when predicting other relevant signals; Removing termination signal does not impact the convergence of reward prediction accuracy or that of the state prediction but the RL performance is hindered. Removing the reward signal or the next state prediction signal leads to a total collapse of the tree-search based behavior policy but the convergence of the remaining model training signals is not affected much. With these, we would like to suggest that we have, at least in this task setting, learned a set-based representation capable of predicting all interesting quantities. We will add the discussions of this matter to the Appendix and we expect to provide in our source code repo the tensorboard data.
>
> We wish to thank you again for your review and hope that our efforts may make you consider accepting our manuscript. Best!

---

> > ### Comment · Reviewer_hG23 · 2021-08-27
> > **Thank you**
> >
> > Thank you for your reply and your hard work; I have updated my review.

---

> > > ### Author Response · Authors · 2021-08-27
> > > **Thank You!**
> > >
> > > Please do not hesitate to reply with further questions. Thank you again for your appreciation!

---

### Official Review · Reviewer_orSE · 2021-07-12

**Rating:** 7
**Confidence:** 4

**Summary:**

The authors introduce an architecture for focusing agents in model-based reinforcement learning on just the relevant subset of the environment representation for making decisions (reward, termination, and successor state prediction). The approach leans heavily on recent advances in set-based representations and the approach is tested in the MiniGrid-BabyAI framework. Results show promising behaviour, particularly in the ability to generalize across different settings.

**Ethical Concerns:**

I see no major ethical concerns with this body of work.

**Limitations And Societal Impact:**

The biggest limitation is the scope of the evaluation. The authors have identified this, along with other key limitations of the approach. Generally, the limitations of the work are carefully considered and discussed.

**Main Review:**

## Originality
The ideas in this paper are novel and interesting. The contrast to related work in the background section is one of the best I've ever read. It is immediately clear what novel aspects are part of this work and the motivation behind them.

## Quality
The submission is technically sound, and the core hypothesis is adequately tested in the evaluation environment. The set of baselines are well-thought-out, and the evaluation seems complete.

The authors are quite honest and clear about the limitations, including the complexity of the hyperparameter space (which is a concern for models such as this).

One concern I have is in the presentation of some of the results. For Fig 7, it looks like the World Models approach is still warming up -- further training would quite possibly change the "best approach". What happens when you allow for a longer budget? I understand that there is a claim to data efficiency, but that is a weaker result than what is currently presented.


## Clarity
I found the paper to be extremely clear and filled with interesting insights -- both on the analysis in the evaluations and on the design decisions compared to related work. Nothing to suggest.


## Significance
This is perhaps the biggest weakness of the paper. I find the ideas compelling, and the paper rides on that + the clarity alone, but the approach is tested in an arguably very simple setting. It is designed to exhibit the phenomena of interest, but there is little indication that the results will generalize to more complex settings. From the difficulty of jointly learning the representation+selection in the bottleneck, to the argument for ignoring reconstruction, there are many decisions that may only serve useful in this one particular environment.


## Questions
I am left with a few open questions after reading the work at various levels:

1. The bottleneck still looks at everything in the memory in order to get the compressed latent state representation. This is arguably different from how we might do it cognitively (going over every fact to find those we might want to change). How might this be improved? It's not only about reasoning on a small set of facts, but avoiding the full $O(n)$ computation of what those $m << n$ facts would be.

2. I am interested in the claim included in the first Background & Context section as to why reconstruction isn't useful. Would this still be the case if given the opportunity for unsupervised pre-training for a decent reconstruction?

3. The "integration" phase appears to not be an inversion of the "selection" phase? If so, why? If they are indeed a mirror (i.e., the integration places the new representation back in the appropriate position), then the paper should clarify this.


**Time Spent Reviewing:**

5

---

> ### Author Response · Authors · 2021-08-10
> **A Point-By-Point Reply to Your Comments**
>
> Thank you very much for your appreciation of our work and such an in-depth inspection, as well as the insightful comments!
>
> We would like to address your concerns point by point:
>
> World model (staged training) baseline would gain a little bit of task performance after the cut-off. However, the seed runs on average would not approach the levels of end-to-end methods. This is possibly because the representations are not tuned for value estimation since the corresponding signal did not participate in the learning during the world model building stage (unsupervised). For your interest, we have done a comparison of the WM baseline against a CP instance with the same architecture, additionally granting WM with a free unsupervised learning phase of 1e6 steps (the figure compares the performance of the two methods after the policy improvement starts, under OOD difficulties 0.25 to 0.55). An additional section regarding this, including a new set of curves and comments, are added to the Appendix. For the figures, please check *fig_free_unsupervised.png* in this anonymous repo: https://bit.ly/3ivcbEC.
>
> As reviewers have suggested, we do recognize the limitations of the current experiment setting and have added new experiments for inspecting the scalability and the applicability on new types of tasks. We hope that this effort would address your biggest concern on the paper (the significance) while we explore the general applicability of the approach. “A newly crafted family of Key-Door tasks is built upon the logic of MiniGrid’s Unlock-v0 and our previous task (with turn-and-forward dynamics). The agent needs to navigate the gridworld while avoiding obstacles (0 reward, end of episode) to get a key first (+0.5 reward) and then unlock a chest (+0.5 reward, end of episode, considered as success) to finish the task. Besides all the other examinations we have done for the previous settings, in this task in particular, the bottleneck is tested against a conditional selection: the reward and termination feedbacks on the chest is conditioned on if the key is still not acquired. Hence, in the bottleneck, when the agent plans to step into the chest, the bottleneck should additionally select the key to predict well the outcome. We prioritize on investigating such effect with the accuracy regarding the selection. The results of the three compared methods are more differentiable than the original turn-or-forward setting and the pattern remains consistent across all settings, the bottleneck-equipped CP agent performs better than UP and modelfree.” For the figures, please check this anonymous repo: https://bit.ly/3ivcbEC.
>
> **For your open questions:**
>
> 1. With a more capable encoder, we expect an agent to only hold in memory the objects related to the task (not necessarily relevant for the particular moment during the task). With this I think our proposition of scanning the objects for interest may seem more adequate.
>
> 2. Reconstruction could be useful while in this work we focus on its potential drawbacks, e.g. weakness to higher-dimensionality and noises. We think that using any bit of information provided by the environment is useful for sample efficiency. We would like to investigate further how to learn a meaningful state representation when reward and termination signals are absent.
>
> 3. The integration phase is indeed intuitively the inverse of the selection phase though not mathematically. We would add clarifications to the paper.
>
> Thank you very much for your kind acceptance of this work and hope that our new efforts may have a positive effect on the rating. Best!

---

> > ### Comment · Reviewer_orSE · 2021-08-13
> > **Rebuttal Response**
> >
> > Thank you for the response -- I think the added evaluations and insight (if only discussed briefly) would contribute a fair bit to the paper.
> >
> > I'd like to expand on your response to Q1 wrt generalization. There may be much about the world that is relevant to only certain tasks, and for cross-task generalization we would hope that our agents learn a representation that allows them to recognize what is relevant. The location of my coffee maker is relevant for the eventual task of having breakfast tomorrow, but the location of my keyboard & mouse are far more relevant to the current task of responding to your comments. Having to scan the total sum of what might become relevant (or has been previously), gives me a linear complexity when we clearly get away with something far more efficient.
> >
> > So if the context of Q1 shifts to the broader notion of an agent that acts in a world with many tasks, and thus the "list of things" grows fairly large, how does this change the feasibility of the approach taken here?

---

> > > ### Author Response · Authors · 2021-08-13
> > > **Reply to the Response**
> > >
> > > We thank you sincerely for your appreciation of this work.
> > >
> > > The new evaluations and insights have already been added to the revised manuscript.
> > >
> > > Your analysis is precise and clearly the proposed was developed for a single-task setting. The current approach needs to scan all the possibly relevant objects provided by the encoder into the unconscious state set. However, we think that when targeting a multi-task setting as you have suggested, task-conditioned computations could be utilized to address the complexity. The agent having its task in mind resembles the other dimension of consciousness (C2), i.e. self-awareness, which is our ongoing research direction following this work.
> > >
> > > The following directions might be promising to reduce the complexity, as we the collaborators have discussed during our brainstorming:
> > >
> > > 1. The encoder could be generative and stochastic, sampling a relevant subset (by opposition to having a deterministic attention which is applied to every possible object). This resembles humans experience, with only a few candidates popping up to our mind, possibly generated by a powerful inference engine, and not always the same candidates being generated.
> > >
> > > 2. The attention over objects could be pre-constrained to a subset conditioned on the task.
> > >
> > > We are very delighted to participate in an in-depth discussion regarding this matter and hence please reply with no hesitation. We hope our efforts for the improvement would have a positive effect on your evaluation of this work.
> > >
> > > Best!
> > >
> > > Authors

---

> > > > ### Comment · Reviewer_orSE · 2021-08-16
> > > > **Final thought**
> > > >
> > > > These are fascinating directions, and while I'd love to dive into them further, it is certainly beyond the central topic of the paper under review -- I don't believe any of these considerations need be explored in the paper. I'll reserve my follow-up for a more appropriate venue.
> > > >
> > > > Thank you for the additional thoughts, and I look forward to seeing where you take this work.

---

> > > > > ### Author Response · Authors · 2021-08-16
> > > > > **Thank You!**
> > > > >
> > > > > Thank you again!
> > > > >
> > > > > Best,
> > > > >
> > > > > Authors

---

### Official Review · Reviewer_dyBZ · 2021-07-16

**Rating:** 7
**Confidence:** 4

**Summary:**

This work presents a model-based reinforcement learning agent that makes use of a bottleneck attention mechanism for the planning module.

**Limitations And Societal Impact:**

Authors state limitations of their study, but make too broad claims for the experiments included as explained above.

**Main Review:**

This paper proposes the use of attention-based bottleneck to improve the performance of model-based RL (MBRL) agents, which is intuitive and well explained. This mechanism seems similar to the top-k attention mechanism presented with BRIMs [1] also targeting OOD generalisation. But to the best of my knowledge, this is the first work that assess the impact of this mechanism in MBRL. Authors also study the impact of using a vector embedding vs am array of vectors (set embedding) which is also interesting. I think these results would be of interest for the RL community.

Clarity and quality is good in general, authors should be commended for the abundant graphs and visual explanations. My comments on this side focus on a few points easily fixable that I raise below. My biggest concern, and the reason why I think the paper cannot be accepted in its current form, are the claims that authors do of the conclusions extracted from their work, which are too generic and broad for the work presented and do not reflect correctly what has been presented:

*Line 304 says that authors have drawn the conclusion that set-based representations are better in multitask environments. I think that if authors want to state this they should use more than one benchmark, currently all the experiments concern the same task which is "reach  green cell while avoiding lava cells"

*Line 307 "model-free methods face difficulties in OOD generalisation". Again too broad when tested on a grid world with a single type of task

*Line 309 states that "online joint learning is good for RL" has the same problem and, moreover this one is a controversial statement since there are evidences in the opposite direction (e.g., [2]), again I think authors cannot do such broad claims with the experiments included.

To clarify, I am not asking the authors to do more experiments to consider accepting this paper, I believe the proposed approach and the experiments included are enough to show the potential of the proposed mechanisms, I am ok if authors just adjust their claims to do more realistic statements according to what they present.

Additional comments:
* Checklist: multiple answers should be expanded, the answer should include brief explanations, most are only "yes"
* Figure 2 caption, it would be helpful to say which appendix section to go for.
* The FC downscale operation, is barely explained, how much they downscale?
*The FC layer in the permutation invariant receives as input an array of vectors, but the FC layer must operate with vectors, I understand that each vector is passed through the FC separately, isn't it?
* Sections 5.3 and 5.4 should be merged.  Section 5.3 is supposed to be about in-distribution evaluation but the last lines are about OOD evaluation, which is section 5.4. I think it is good to have the text as it is now, thus, I would only put everything in the same subsection.
* Lines 111-115. You cite twice "[10]" but seems tha tehy should be different references (one to refer what are you following and the other to refer to other families of approaches).
*Caption Figure 10, you say " Note that the concatenation does not happen outside the residual pass in case the dimensions do not match" but from the picture is  looks like in the residual connection is where there is no concatenation of actions.

--After Rebuttal---
Authors have correctly addressed my concerns about the claims and updated them accordingly. This, together with the additional experiments and updates makes me confident to raise my score and recommend the acceptance of this work.

[1] Mittal, Sarthak, et al. "Learning to combine top-down and bottom-up signals in recurrent neural networks with attention over modules." International Conference on Machine Learning. PMLR, 2020.
[2] Lehuger, Auguste, and Matthew Crosby. "Fixed $\beta $-VAE Encoding for Curious Exploration in Complex 3D Environments." arXiv preprint arXiv:2105.08568 (2021).

**Time Spent Reviewing:**

7

---

> ### Author Response · Authors · 2021-08-10
> **A Point-By-Point Reply to Your Comments**
>
> Thanks for your appreciation!
>
> We would like to address your concerns point by point:
>
> For your suggestions regarding Line 304, 307 and 309, we do see the necessity of limiting our claims and moderating our conclusions and we sincerely thank you for the suggestion. We have moderated our statements and **have fixed such problem across the whole manuscript**. Since we are adding more experimental settings following the reviewers’ suggestions, we would limit our comments and conclusions to the extended cases of experiments.
>
> Here is a brief description of the added experimental content: *“A newly crafted family of Key-Door tasks is built upon the logic of MiniGrid’s Unlock-v0 and our previous task (with turn-and-forward dynamics). The agent needs to navigate the gridworld while avoiding obstacles (0 reward, end of episode) to get a key first (+0.5 reward) and then unlock a chest (+0.5 reward, end of episode, considered as success) to finish the task. Besides all the other examinations we have done for the previous settings, in this task in particular, the bottleneck is tested against a conditional selection: the reward and termination feedbacks on the chest is conditioned on if the key is still not acquired. Hence, in the bottleneck, when the agent plans to step into the chest, the bottleneck should additionally select the key to predict well the outcome. We prioritize on investigating such effect with the accuracy regarding the selection. The results of the three compared methods are more differentiated than the original turn-or-forward setting and the pattern remains consistent across all settings, the bottleneck-equipped CP agent performs better than UP and modelfree.”* For the figures, please check this anonymous repo: https://bit.ly/3ivcbEC.
>
>
> **For your additional comments**:
>
> -- Checklist has been expanded with brief explanations.
>
> -- Appendix references and links have been provided.
>
> -- FC downscale is a linear transformation which downscales the dimensionality of the intermediate objects to that of the features part of objects (before layer-normed). In this way, after concatenating the positional tails the objects have consistent dimensionality.
>
> -- FC does indeed accept the object vectors one-by-one however the implementation parallelizes the whole bundles of such operations.
>
> -- Section 5.3 and 5.4 are merged as you suggested, with wording improvements.
>
> -- On line 111-115, [10] proposes an object detection method which contains parts similar and different to our approach. Commonality: object features are chopped CNN feature maps; Difference: positional embeddings are added as other methods in [10] however we concatenate for the sake of alignment, as discussed in the manuscript. We would change the citations to avoid confusion.
>
> -- On action concatenation of the residual pass: we facilitate X^'=X+f(cat[X,a]), where X is the set of objects input to the FC part of the action-conditioned transformer layer, cat([X,a]]) is the concatenation of action embedding a to every object emdedding in X and X^' is the output set. Note that f downscales the dimensionality of its input to match X. The confusing wording has been fixed, and this very piece of clarification is added to the manuscript.
>
> We wish to thank you again for your review and hope that our efforts may make you consider accepting our manuscript. Best!

---

> > ### Comment · Reviewer_dyBZ · 2021-08-15
> > **Follow up**
> >
> > Thank you for your response and clarifications.
> >
> > Could you please put in a further response the content of the updated summary of old section 5.5?
> >
> > Best,
> > dyBZ

---

> > > ### Author Response · Authors · 2021-08-15
> > > **To Your Follow Up**
> > >
> > > Thank you very much for reading through our comments!
> > >
> > > Here is the updated (sub-)section (changed from 5.5 to 5.4 due to that old 5.3 and 5.4 are merged)
> > >
> > > *Summary of Experimental Results*
> > >
> > > *With the scope limited to our case of experiments, the results allow us to draw these conclusions:*
> > >
> > > -- *Set-based representations enable at least in-distribution generalization across different environment instances in our non-static setting, where the learners are forced to discover dynamics that are preserved across environments;*
> > >
> > > -- *Model-free methods seem to face more difficulties in solving our OOD generalization tasks which preserved the same environment dynamics to the corresponding in-distribution training settings;*
> > >
> > > -- *MPC exhibits better performance than Dyna-style planning in the tested OOD generalization settings;*
> > >
> > > -- *Online joint training of the representation with all the relevant signals could bring benefits to RL, as suggested in [25]. Please check Appendix E for more discussions of this matter;*
> > >
> > > -- *In accordance with our intuition, transition models with bottlenecks tend to learn dynamics better in our tests. This is likely for they prioritize learning the relevant aspects, while models without bottleneck may have to waste capacity on irrelevance;*
> > >
> > > -- *From further experiments provided in the Appendix E, we observe that bottleneck-equipped agents are also less affected by cluttering and larger sets of objects in a state,  possibly due to their prioritized learning of interesting entities.*
> > >
> > > Please do not hesitate to reply to point out if there is anything that you think we could work on to improve. We hope our efforts for the improvement would have a positive effect on your evaluation of this work.
> > >
> > > Best,
> > >
> > > Authors

---

> > > > ### Comment · Reviewer_dyBZ · 2021-08-22
> > > > **Response**
> > > >
> > > > Thank you for the response. I am happy to increase my score reflecting your updates to the work.

---

> > > > > ### Author Response · Authors · 2021-08-22
> > > > > **Thank You**
> > > > >
> > > > > Thank you very much for the confidence in this work!

---

### Official Review · Reviewer_sQ3E · 2021-07-24

**Rating:** 6
**Confidence:** 5

**Summary:**

The authors introduce a model-based deep reinforcement learning algorithm that uses tree search based MPC over a learned latent space. In this work, the latent space does not use a vectorized representation but rather a set-based representation with a small bottleneck. The authors then compare their methods to baselines and ablations on a toy environment.

**Limitations And Societal Impact:**

This paper does not add limitations or potential negative societal impact to existing reinforcement learning methods.

**Main Review:**

I want to thank the authors for their work. The paper is rather clear, well structured and easy to follow. However, there are many abbreviations in the experimental part that I believe may be unusual for most readers, which makes it hard to follow.
I think that this research domain is very promising. Indeed, MBRL methods recently demonstrated impressive results and studying how to better structure latent spaces is an exciting research direction. However, this paper exhibits many limitations:

This method falls in the category of deep MBRL methods, however it lacks many references to modern state-of-the-art methods in the field. Notably, the authors mention Line 30 that except for MuZero, MBRL methods show poor performance. They also argue that only Predictron and MuZero plan over a latent space while the others methods construct a model over the true state space. I think that the authors missed many recent and important references. Notably, methods such as PETS, MBPO, Dreamer or LEAP demonstrated impressive results in terms of both sample efficiency and asymptotic performance on challenging benchmarks. Among these methods both LEAP and Dreamer plan over a latent space. Furthermore, Dreamer also introduced set-based representation. Thus, it would be key in this paper to cite Dreamer and to carefully position the proposed method with respect to it.

As this paper does not provide theoretical results or analysis, the experimental section is key to demonstrate the usefulness of the method. However, this section is in my sense too weak to achieve this goal. Indeed, the authors considered a single toy environment that does not exhibit any of the challenges that the deep RL and deep MBRL have been designed to address. I would highly recommend the authors to assess the performance of their method on standard benchmarks such as the Atari suite or the Gym Mujoco environments. In addition, the authors do not compare their methods to classical SOTA methods such as the one aforementioned. The authors should compare their algorithm to at least Dreamer.

Regarding the paragraph between lines 72 and 83, the authors missed the reference to Hamrick et al. (on the role of planning in model-based deep reinforcement learning) which I believe is important to mention here.

The conclusions in Section 5.5 are too broad and in my opinion, not enough justified. For instance, the authors claim that "Model-free methods face difficulties in OOD generalization;" where Model-free methods refer to double DQN and this method has been tested on a single toy environment. There have been tremendous efforts to study and improve the generalization of model free methods and I would recommend the authors either moderate their conclusions or to better inform them with more experiments and relevant citations.

Regarding the notations, it is misleading to use $\mathcal{S}$ for the latent state space and $\hat{s}$ for latent states. Most of the time, $\hat{s}$ is used for reconstructed states. I would recommend another notation such as $z$ for latent variables. Also, when the authors introduce the action space line 44, I would recommend to mention whether these actions are assumed to be discrete or continuous.
To conclude, I think that this research direction is promising, however, this paper lacks many important references and exhibits too weak an experimental study. I would recommend the authors strengthen their positioning with respect to the existing literature and either bring theoretical elements or consider more challenging environments and baselines in their study.


**Time Spent Reviewing:**

4

---

> ### Author Response · Authors · 2021-08-10
> **A Point-By-Point Reply to Your Comments**
>
> Thank you! We would like to start by rebutting some of your major concerns and then with how we improved the work based on your helpful feedback.
>
> **PART I: Rebuttal**
>
> Before diving into the details, we would like to clarify that this work targets a quite new topic of RL: “improving OOD generalization capabilities for planning using a bottleneck” instead of “(in-distribution) generalization capabilities”, “sample efficiency for solving tasks” or “studying how to better structure latent spaces for MBRL”. Such purpose was investigated with a minimalistic MBRL architecture on suitable settings, each with a demanding dynamic randomized episode generation. All the comments and conclusions are developed around the OOD capabilities brought by the bottleneck during planning. In other words, we do not focus on achieving certain performance on certain set of tasks with some detailed design of deep architectures but on providing a set of controlled experiment to verify our hypotheses.
>
> Hence, with utmost respect, we want to politely suggest that your claim “the authors considered a single toy environment that does not exhibit any of the challenges that the deep RL and deep MBRL have been designed to address” may be based on an inaccurate understanding of the scope of this work. Such misunderstanding could be an accidental result of our wording. To help you better understand our paper, we have tried our best to improve during this response window.
>
> Additionally, we used DRL and MBRL to investigate the novel direction of OOD generalization and cognition inspired inductive biases. We believe that this direction is interesting and promising as DRL is not only about learning representations from an environment with complex characteristics, such as image-based Atari.
>
> Please allow us to introduce the reasons why we have adopted the current experimental setting, which gave you the seemingly unexpected impression of a “toy” set. We found that, at least, the following features are required for suitable experiments to investigate the characteristics of the bottleneck (After this revision, we have made sure that these are stated more clearly in the manuscript):
>
> 1. Well-defined objects, object-interaction based dynamics. This is the key to the assumption of the paper’s applicable scenarios. No irrelevant challenges regarding OOD generalization should be posed to the agent, such as exploration in NetHack NLE or encoding objects from image-based observations such as in Atari and ProcGen (which we leave for future work).
>
> 2. Controllability over the OOD evaluation setting. It is desirable that the optimal policies of the tasks are solvable so that we are able to measure the quality of planning during OOD tests.
>
> 3. The environment should be discrete with deterministic dynamics. This is a setting where tree-search based planning shows stable behavior. For this reason, we would hesitate to test the idea on a mujoco environment for that tree-search in a continuous-dynamics environment is not of the paper’s concern and remains as an open topic.
>
> Even Gym-MiniGrid was only able to offer a programming interface for us to implement a relatively satisfactory setting. We refined this environment over iterations to fit our goal to build a suitable testbed. Limited to this scope, we have indeed tried our best to provide a very detailed and insightful investigation to the proposed method. Here is a summary of our experimental settings after this revision (in the table, only the new or updated contents are highlighted with **bold font**):
>
> |Nickname	|DistShift-v1	|DistractedShift-v3	| **KeyDoor-v3** |
> |---|---|---|---|
> |Location in File	|Main Manuscript, Experiment Section	|Appendix E	| **New, added to Appendix E** |
> |World Size	| **6x6 to 10x10**	|8x8	|8x8|
> |Agent Dynamics	|Turn-OR-Forward (turn left, right, or forward)	|Turn-AND-Forward (turn-left-forward, …, turn-back-forward)	| **Turn-AND-Forward** |
> |Additional Configuration	 |	|Color distraction added for inspecting the bottleneck’s robustness against cluttering	| **A key must be fetched before unlocking a chest. With lava pools spawned for extra difficulty.** |
> |Compared Methods	|All (**Dreamer's performance is evaluated here**)	|CP, UP, modelfree	|CP, UP, modelfree|
> |#runs	|**20**	|**20**	|**20**|
>
> We sincerely suggest that you reconsider this work while we gladly accept your other helpful suggestions and comments, as in the following part of the reply.
>
> **PART II: Improvements**
>
> On experiments (for important updated figures please check this anonymous link: https://bit.ly/3ivcbEC.):
>
> We propose an alternative solution which shows additional applicability of the methods and satisfy our experimental requirements on OOD generalization: we added another set of experiments with different dynamics to diversify the experiments. Here is a list of the new experiments and the brief results:
>
> 1. Extended results on different sizes of gridworlds on top of the original set of experiments (with both turn-and-forward and turn-or-forward action spaces). We provide additional results on from 6x6 to 10x10 world sizes. The conclusions regarding the bottleneck are consistent. Based on our observations, world sizes smaller than 6x6 would pose too simple a challenge for differentiated performance (and chances for memorization) while starting from 10x10 the difficulty of exploration kicks in (the variance in the curves becomes large) and the computational requirements become very harsh.
>
> 2. A newly crafted family of Key-Door tasks is built upon the logic of MiniGrid’s Unlock-v0 and our previous task (with turn-and-forward dynamics). The agent needs to navigate the gridworld while avoiding obstacles (0 reward, end of episode) to get a key first (+0.5 reward) and then unlock a chest (+0.5 reward, end of episode, considered as success) to finish the task. Besides all the other examinations we have done for the previous settings, in this task in particular, the bottleneck is tested against a conditional selection: the reward and termination feedbacks on the chest is conditioned on if the key is still not acquired. Hence, in the bottleneck, when the agent plans to step into the chest, the bottleneck should additionally select the key to predict well the outcome. We prioritize on investigating such effect with the accuracy regarding the selection. The results of the three compared methods are more differentiated than the original turn-or-forward setting and the pattern remains consistent across all settings, the bottleneck-equipped CP agent performs better than UP and modelfree.
>
> On comparison with Dreamer and other methods:
>
> While we should have highlighted the feats achieved by methods such as PETS, MBPO, Dreamer or LEAP, which we will certainly include in the revision, we cited MuZero for the purpose of differentiating the methods’ representation learning methodologies (of the states): a state representation which only targets the value estimation on a single task may encounter difficulties in an OOD setting.
>
> We would also like to point out that this work took approximately 2 years and during the course of the study we do not have any knowledge of the method (Dreamer) as the works are concurring. Here are the efforts we have made adapting DreamerV2 to the experiments and the corresponding results, to be added to the manuscript and source code (both tensorflowV2).
>
> 1. Choice of encoder: either we flatten the extracted feature map to match that of the vectorized DreamerV2 features or get it consistent to set-based as in our setting. For the latter, since we are targeting deterministic environments, we can dodge the difficulties of stochastic set generation using our designed model. Based on DreamerV2 source code, we refer to the agent using the same no-set vectorized encoder from the baseline in our work but everything else theirs “DreamerV2-noset” and the agent using our set-based encoder with deterministic model “DreamerV2-deterministic”.
>
> 2. “DreamerV2-noset” fails with very low overall success rate on both in-dist and OOD tasks. Two reasons are speculated: 1) the incapability of memorization using the vectorized states in our non-static setting and 2) undesirable decision-time model-free behavior which is discussed for Dyna. The results do not improve after some hyperparameter and architecture tuning.
>
> 3. “DreamerV2-deterministic” does not learn well within in-dist tasks possibly due to the delusional value updates brought by the imperfectly learned models. It fails on OOD tests too, despite that the UP model learns well according to the accuracy in state feature, reward and termination predictions. During OOD test decision-time, Dreamer behaves model-free based on value estimator alone.
>
> On minor concerns:
>
> 1. We will cite [Hamrick et al.] as you have pointed out its importance to the topic of the paper.
>
> 2. We will moderate our conclusions of Section 5.5 by adding more comments comparing to the recent work in DRL.
>
> 3. We will adopt a separate set of notations to differentiate the learned state space and the true state space.
>
> Thank you very much for patiently reading through the reply! We sincerely hope that you would consider accepting this work for our new efforts! Best!

---

> > ### Comment · Reviewer_sQ3E · 2021-08-26
> > **Feedback to your point by point reply**
> >
> > Thank you very much for your detailed point by point reply, it actually answered my concerns and as a consequence I will raise my score for the paper. I still think it would be very interesting to see how the method works on more involved benchmarks and how it compares to other recent methods. For example, it could be really interesting to compare it to methods such as RIDE (Raileanu, et al. 2019) or AGAC (Flet-Berliac et al. 2020) that consider procedurally generated environments and also learn some models and compare it on the same benchmarks that are considered. This being said I'm happy with your response, thanks a lot for taking my feedback into account.

---

> > > ### Author Response · Authors · 2021-08-26
> > > **Thank You!**
> > >
> > > We are extremely happy that you are content with our reply! Thank you very much for your support and if you have any questions do not hesitate to leave your comments!
> > > Thank you again! We await your adjustment to the rating.
> > >
> > > Best,
> > > Authors

---

### Decision · Program_Chairs · 2021-09-27

**Decision:**

Accept (Poster)

**Comment:**

The reviewers agreed that the paper contains compelling ideas and should be accepted.  Some initial concerns about the generality of the claims were well addressed by the authors.